# Repositioning of Old Drugs for Novel Cancer Therapies: Continuous Therapeutic Perfusion of Aspirin and Oseltamivir Phosphate with Gemcitabine Treatment Disables Tumor Progression, Chemoresistance, and Metastases

**DOI:** 10.3390/cancers14153595

**Published:** 2022-07-23

**Authors:** Bessi Qorri, Reza Bayat Mokhtari, William W. Harless, Myron R. Szewczuk

**Affiliations:** 1Department of Biomedical and Molecular Sciences, Queen’s University, Kingston, ON K7L 3N6, Canada; bessi.qorri@queensu.ca (B.Q.); reza.bayatmokhtari@sickkids.ca (R.B.M.); 2ENCYT Technologies Inc., Membertou, NS B1S 0H1, Canada

**Keywords:** osmotic pump, chemoresistance, drug repurposing, EMT

## Abstract

**Simple Summary:**

Repositioning old drugs in combination with clinical standard chemotherapeutics opens a promising clinical treatment approach for patients with pancreatic cancer. This report presents a therapeutic repositioning of continuous perfusion of aspirin and oseltamivir phosphate in combination with gemcitabine treatment as an effective treatment option for pancreatic cancer. The data suggest that repositioning these drugs with continuous perfusion with gemcitabine disables chemoresistance, tumor progression, EMT program, cancer stem cells, and metastases in a preclinical mouse model of human pancreatic cancer. These promising results warrant additional investigation to assess the potential of translating into the clinical setting to improve the cancer patient prognosis for an otherwise fatal disease.

**Abstract:**

Metastatic pancreatic cancer has an invariably fatal outcome, with an estimated median progression-free survival of approximately six months employing our best combination chemotherapeutic regimens. Once drug resistance develops, manifested by increased primary tumor size and new and growing metastases, patients often die rapidly from their disease. Emerging evidence indicates that chemotherapy may contribute to the development of drug resistance through the upregulation of epithelial–mesenchymal transition (EMT) pathways and subsequent cancer stem cell (CSC) enrichment. Neuraminidase-1 (Neu-1) regulates the activation of several receptor tyrosine kinases implicated in EMT induction, angiogenesis, and cellular proliferation. Here, continuous therapeutic targeting of Neu-1 using parenteral perfusion of oseltamivir phosphate (OP) and aspirin (ASA) with gemcitabine (GEM) treatment significantly disrupts tumor progression, critical compensatory signaling mechanisms, EMT program, CSC, and metastases in a preclinical mouse model of human pancreatic cancer. ASA- and OP-treated xenotumors significantly inhibited the metastatic potential when transferred into animals.

## 1. Introduction

The molecular pathogeneses and new therapeutic targets with a focus on pancreatic cancer have been eloquently reviewed by Wong and Lemoine [1,2], Maitra and Hruban [3], and Javadrashid et al. [4]. Here, pancreatic tumorigenesis is affected by only a few critical signaling pathways affected by genetic alterations. For example, the essential signaling pathways, such as epidermal growth factor receptor (EGFR), vascular endothelial growth factor (VEGF), gastrin hormone, matrix metalloproteinase (MMP), and rat sarcoma (Ras) have been targeted with clinical therapeutic intent, but these clinical treatments have been discouraging. For example, the failures of (a) Bevacizumab, a humanized VEGF antibody with gemcitabine and erlotinib, (b) Sorafenib, a multi-targeted kinase inhibitor of VEGF receptors, platelet-derived growth factor receptors (PDGFR), stem cell factor receptor/c-Kit, Raf-1 protooncogene, serine/threonine kinase (RAF1) and Fms-like tyrosine kinase-3 (FLT-3), and (c) Axitinib, an orally active VEGFR inhibitor and related tyrosine kinase receptors, collectively, have proven to be less efficacious in specifically targeting and killing of cancer cells [2].

Intrinsic resistance of pancreatic cancer cells to chemotherapy treatment is well recognized clinically and, in the laboratory [5,6,7,8,9]. However, combining first-line chemotherapies in treating metastatic pancreatic cancer, including Abraxane with gemcitabine (GEM) or 5-FU with Oxaliplatin and Irinotecan, can control the disease process for a median length of time of approximately six months in patients studied [3,4,10,11]. An estimated 20–30% of patients will experience a response with these treatments manifested by shrinkage of the primary tumor and metastases. However, acquired treatment resistance inevitably develops, and when this happens, patients often die rapidly from their diseases. Given its high lethality and time-limited response to current treatments because of intrinsic and acquired resistance, there is an urgent unmet need for novel treatments for pancreatic cancer.

While there are competing theories as to why resistance develops to chemotherapy treatment, an emerging theory with empirical support is that cancer treatments such as chemotherapy may foster treatment resistance through upregulation of the epithelial–mesenchymal transition (EMT) program [12,13,14,15,16]. EMT is a highly conserved developmental de-differentiation program whereby a polarized epithelial cell undergoes distinct biochemical changes to enable it to assume a mesenchymal phenotype associated with the ability to migrate, invade tissue, and resist apoptosis [17,18]. Intriguingly, more differentiated cancer cells experimentally induced to undergo the EMT process can acquire properties of stem cells [19]. Cancer stem cells (CSC) are not only more resistant to chemotherapy treatments, but they are also very effective at promoting angiogenesis and micro-metastases formation [20]. During the EMT de-differentiation program, more differentiated epithelial cells can acquire properties of stem cells. Experimental induction of EMT in cancer cells was found to promote invasion, metastasis, and resistance to chemotherapy [21]. For example, cancer cells that have undergone the EMT process, even a partial EMT, can see an increase in the half-maximal inhibitory concentration (IC50) dose of a chemotherapy drug gemcitabine by ∼10-fold. For example, O’Shea et al. [22] reported that oseltamivir phosphate (OP) upended the chemoresistance of PANC-1 to cisplatin and gemcitabine alone or in combination in a dose-dependent manner. Additionally, OP reversed the EMT characteristic of E-cadherin to N-cadherin changes associated with resistance to cisplatin and gemcitabine therapy. Notably, the epidermal growth factor receptor (EGFR) is critical in inducing the EMT program in pancreatic cancer [23].

The relationship between EMT and metastasis is poorly understood because the PDAC metastatic genetic composition resembles the complementary primary tumors [24,25,26]. The promoters of metastasis in PDAC are TP53 [27], SMAD4 [28], Wnt [29], and ECM gene expression [30]. The transforming growth factor beta (TGF-β) has been reported to induce tumor cell invasion and migration following the initiation of EMT [31,32].

Previous reports show that OP can limit tumor growth, neovascularization, and metastasis of MDA-MB-231 human triple-negative breast cancer cells [33] and MiaPaCa-2 [34] pancreatic cancer cell lines in tumor xenografts in RAGxCγ double mutant mice. The findings also revealed a novel signaling platform regulating EGFRs [34,35]. Additionally, we recently discovered that aspirin (ASA) and celecoxib can specifically target and inhibit Neu-1 sialidase activity in EGF-induced receptor activation in PANC-1 cells and their GEM-resistant variant and MiaPaCa-2 pancreatic cancer cells [7]. The graphical abstract here illustrates that mammalian Neu-1 forms a complex with matrix metallopeptidase 9 (MMP-9) and G protein-coupled receptor (GPCR) neuromedin, involved in the regulation and activation of several receptor tyrosine kinases (RTKs), such as EGFR [34], the insulin receptor (IR) [36], and the nerve growth factor (NGF) TrkA receptor [37], and TOLL-like receptors (TLRs) [38,39,40,41]. All of these receptors are activated in cancer cells. When the ligand binds to its receptor, the receptor undergoes a conformational change which results in the activation of MMP-9 via Gαi subunit signaling in order to remove elastin binding protein (EBP). The removal of EBP activates Neu-1 in complex with the protective protein cathepsin A (PPCA) [42]. Activated Neu-1 in complex with the receptor at the ectodomain hydrolyzes terminal α-2,3-sialyl residues to remove steric hindrance for receptor dimerization and downstream signaling pathways [34].

Targeting single oncogenic pathways may not be a promising pancreatic cancer treatment. Future studies must suppress pancreatic tumor cells’ multiple enabling hallmark(s) capabilities. Despite the therapeutic interventions, the cancer cell survival program is adaptive and invasive, so more aggressive phenotypes will survive and metastasize. The clinical evidence is that repeat instability emerges first, followed by more significant aberrations, with compensatory effects leading to robust tumor fitness maintained throughout the progression. In the present study, we identified a novel treatment approach targeting Neu-1 that is hypothesized to ablate these compensatory effects of pancreatic cancer. Here, a continuous infusion of OP combined with ASA is used with standard treatment with GEM in a preclinical mouse model of human pancreatic cancer. When taken orally, OP is converted through first-pass metabolism in the liver to oseltamivir carboxylate, which is ineffective against mammalian Neu-1 [40]. To avoid this first-pass metabolism to the ineffective carboxylate form of the drug, we administered OP as a continuous subcutaneous infusion via an osmotic pump surgically implanted at the tumor site. We also incorporated ASA based on our previous work showing that ASA specifically targeted and inhibited Neu-1 sialidase activity [7]. Notably, Neu1 regulates EGF-induced receptor activation in pancreatic PANC-1 cancer cells and its GEM-resistant variant, PANC-1-GemR cells [6,7].

Osmotic pumps can deliver drugs providing sustained, controlled, and slow release of the drugs for up to 42 days. We hypothesized that the continuous infusion of OP with ASA might effectively disrupt the development of acquired chemoresistance during chemotherapy treatment due to their ability to target several receptor tyrosine kinases implicated in the EMT process. Here, the data indicate that implanted osmotic pumps providing a continuous infusion of OP and ASA with GEM therapy impeded tumor growth, chemoresistance, and metastasis in tumor xenografts in a mouse model of human pancreatic cancer. The tumorigenic and metastatic potential of the xenotumors from the animals treated with the experimental protocols were significantly ablated when transferred into the mammary fat pads of NSG (NOD SCID gamma) branded mice.

## 2. Materials and Methods

### 2.1. Cell Line and Culture

Human pancreatic cancer cell line PANC-1 (ATCC CRL-1469) was obtained from the American Type Culture Collection (ATCC, Manassas, VA, USA) and maintained in Dulbecco’s Modified Eagle’s Medium (DMEM’ Gibco) supplemented with 10% fetal bovine serum (FBS; HyClone), and 0.5 µg/mL Plasmocin (InvivoGen, San Diego, CA, USA). Cultured cells were incubated in 5% CO_2_ at 37 °C.

### 2.2. Reagents

Acetylsalicylic acid (>99% pure, Sigma-Aldrich, Steinheim, Germany) was dissolved in dimethyl sulfoxide (DMSO) to prepare a 550 mM stock solution, which was stored in aliquots at −20 °C. The highest aspirin concentration contains less than 0.5% *v/v* of DMSO in 1 × PBS at a pH of 7. Oseltamivir phosphate-USP (batch # MBAS20014A, >99% pure powder, Solara Active Pharma Sciences Ltd., New Mangalore-575011, Karnataka, India) was freshly dissolved in sterile 0.9% normal saline before use. Gemcitabine hydrochloride (Sigma-Aldrich Canada Ltd., Oakville, ON, Canada) was dissolved in 1 × PBS to create a 133.5 mM gemcitabine stock. This stock was diluted into 0.2 mL sterile 0.9% normal saline for animal injection preparation.

### 2.3. Heterotopic Xenograft Mouse Model of Human Pancreatic Cancer

An immunodeficient mouse model with a double mutation combining recombinase activating gene-2 (RAG2) and common cytokine receptor γ chain (Cγ) was used as xenograft mice. The RAG2 x Cγ double mutant mice on a BALB/c genetic background are completely alymphoid (T-cell, B-cell, and NK-cell deficient), show no spontaneous tumor formation, and exhibit normal hematopoietic parameters. Mice were generated by inter-crossing and were maintained in specific-pathogen-free (SPF) isolators in the Animal Care Facility, Queen’s University, Kingston, Ontario K7L3N6, Canada. A colony was established in the animal facility. All mice were kept under sterile conditions in micro-isolators or air-filtered cages and were provided with autoclaved food and water. All mice used in the studies were approved by the Animal Care Committee, Queen’s University. Female mice between 6 and 8 weeks of age and an average weight of 30 g were used. Animals were anesthetized via inhalation of isoflurane before all surgical procedures and observed until fully recovered. Animals were sacrificed by cervical dislocation. Mouse weight and tumor volumes were recorded for 50 days, after which animals were sacrificed and necropsied.

### 2.4. Cancer Cell Implantation in RAGxCγ Double Mutant Xenograft Mice

The PANC-1 cell line was grown in a 75 cm^2^ cell culture flask at 80% confluence. The cells were resuspended into a solution using TrypLE Express (Thermo Fisher Scientific Canada, Mississauga, ON, Canada) and washed with sterile 0.9% normal saline. After the cell suspension was centrifuged for 5 min at 900 rpm, the cell pellet was suspended in sterile saline at a concentration of 5–10 × 10^6^ cells/mL for 1 × 10^6^ cell implantation subcutaneously into the right-back flank of the mouse. Tumor measurements were taken three times a week, and tumor volumes were determined by (width square/2) × length. At the experiment endpoint, mice were euthanized by cervical dislocation and live necropsy, liver, lung, and tumor weights, fixed in buffered formaldehyde and embedded in paraffin with hematoxylin and eosin (H&E) staining of tissues. Spleen and pancreas were also weighed at necropsy.

### 2.5. Surgical Implantation of ALZET Mini Pump Containing OP and ASA in RAGxCγ Double Mutant Xenograft Mice

The osmotic treatment began when the tumor was palpable at 100 mm^3^, and the tumor volume reached approximately 100 mm^3^. Animals were treated weekly with 100 mg/kg Gemcitabine (IP). Animals receiving drug treatments received OP 40 mg/kg/day plus ASA 50 mg/kg/day for a total of 42 days via a mini-osmotic pump (ALZET Osmotic Pumps, 10260 Bubb Road, Cupertino, CA 95014-4166, USA). These pumps (internal volume, 200 uL) continuously deliver test agents at a rate of 0.15 µL/h for daily drug dosages for 42 days. The ALZET pump’s rate of delivery is controlled by the water permeability of the pump’s outer membrane. The delivery profile of the pump is independent of the drug formulation dispensed. Drugs of various molecular configurations, including ionized drugs and macromolecules, can be dispensed continuously in various compatible vehicles at controlled rates. The molecular weight of a compound, or its physical and chemical properties, has no bearing on its delivery rate by ALZET pumps. While the volume delivery rate of the pump is fixed, different dosing rates can be achieved by varying the concentration of the agent in the solution or suspension used to fill the pump reservoir. The pump was surgically implanted subcutaneously near the tumor under sterile conditions after a small incision. The incision was closed with clap sutures.

### 2.6. Hematoxylin and Eosin (H&E) Staining

Fixed tumors were embedded in paraffin, sectioned at 5 µm, and sections were transferred onto glass slides, deparaffinized through xylene and graded alcohols into the water, and stained with hematoxylin and eosin (H&E). Slides were dehydrated and mounted. All stained sections were assessed on a light microscope (Nikon Eclipse SE EI R STG HNDL TRINOC-oil obj set with digital sight 1000 microscope camera).

### 2.7. Immunohistochemistry (IHC)

Fixed tumors were embedded in paraffin and sectioned at 5 µm. Sections were transferred onto glass slides, deparaffinized through xylene, and graded alcohols into water. For routine immunohistochemical (immunoperoxidase) labeling, antigen retrieval was performed in 10 mM sodium citrate buffer (pH 6) by heating in a microwave oven for 10 min. The sections were cooled for 20 min to room temperature and then incubated in 3% hydrogen peroxide in water for 10 min to block endogenous peroxidase activity. The sections were washed with deionized water for 5 min 10× and incubated with 10% goat or rabbit serum in PBST for 30 min to block non-specific binding.

The primary antibody was diluted in 5% BSA-PBST (1:100–1:500), added to the tissue sections, and incubated overnight at 4 °C. The primary antibodies were obtained from Santa Cruz Biotechnology: pancreatic cancer marker (pan-cytokeratin), differentiation marker (PDX-1), proliferation marker (Ki67), E-cadherin (sc8426), N-cadherin (sc-8424), HIF-1α (ab1; Abcam Inc., c/o 913860, PO Box 4090 Stn A, Toronto, ON M5W 0E9, Canada), HIF-2α (ab109616; Abcam Inc., Tokyo, Japan), CAIX (ab184006; Abcam Inc., Tokyo, Japan), mTOR (ab109268; Abcam Inc., Tokyo, Japan), PI3K (ab140307; Abcam Inc., Tokyo, Japan), and Akt1 (ab8805; Abcam Inc., Tokyo, Japan), hypoxia markers (HIF-1α, HIF-2α, CAIX, pimonidazole), stem cell markers (CD24, CD44, CD36, and CXCR4), and EMT markers (E-cadherin, N-cadherin, EpCAM, and vimentin). Antibodies were made up of 5% BSA/PBST and incubated at 4 °C overnight. The sections were washed with PBS 10× for 5 min each, incubated with the appropriate secondary antibodies, and incubated for 2 min with DAB (3,3′-diaminobenzidine, Vector Laboratories, Inc., Burlingame, CA, USA). The sections were washed with deionized water and counterstained with hematoxylin. Once the counterstaining was complete, the slides were dehydrated and mounted. All stained sections were assessed on a light microscope (Nikon Eclipse SE EI R STG HNDL TRINOC-oil obj set with digital sight 1000 microscope camera).

Pimonidazole, an effective and nontoxic exogenous 2-nitroimidazole hypoxia marker, forms adducts with thiol groups in proteins, peptides, and amino acids. To detect hypoxic cells in tumors, mice were injected intraperitoneally with 60 mg kg^−1^ pimonidazole hydrochloride (Hypoxyprobe-1, NPI Inc., Belmont, MA, USA) from a stock solution containing 20 mg/ mL in sterile PBS, 1.5 h prior to euthanasia. Typically, the tumor was fixed and paraffin-embedded. Tissue sections were deparaffinized through xylene and graded alcohols into water. Sections were antigen retrieved in 10 mM sodium citrate buffer (pH 6.0) by pressure heating in a microwave oven for 10 min. After cooling to room temperature (20 min), non-specific binding was blocked by incubation in 4% BSA/PBS for 10 min. Sections were subsequently incubated overnight at 4 °C with anti-pimonidazole (MAb1, 1:50) (Hypoxyprobe, Inc.121 Middlesex Turnpike, Burlington, MA, USA) followed by secondary antibody conjugated with HRP and incubated for 2 min with DAB (3,3′-diaminobenzidine, Vector Laboratories, Inc., Burlingame, CA, USA). The sections were washed with deionized water and counterstained with hematoxylin. Once the counterstaining was complete, the slides were dehydrated and mounted. All stained sections were assessed on a light microscope (Nikon Eclipse SE EI R STG HNDL TRINOC-oil obj set with digital sight 1000 microscope camera). Sufficient concentrations of pimonidazole adducts are formed on the surface of cells to elicit a response to hydroxyprobe: pimonidazole.

### 2.8. TUNEL Assay

The terminal deoxynucleotidyl transferase dUTP nick end labeling (TUNEL) assay was performed on 5 µm sections prepared from formalin-fixed, paraffin-embedded xenografts using the HRP-DAB TUNEL Assay Kit (ab206386). Paraffin-embedded tissue sections were rehydrated in xylene and graded alcohols and permeabilized following the manufacturer’s procedures. Endogenous peroxidases were inactivated with 3% hydrogen peroxide in methanol. Slides were equilibrated with TdT equilibration buffer. Slides were labeled with the TdT enzyme in the TdT labeling reaction mix for 1.5 h. The reaction was stopped with the stop buffer for 5 min. The conjugate was added to slides for detection. The DAB solution was used for 15 min for development. Slides were counterstained with hematoxylin. Once the counterstaining was complete, the slides were dehydrated and mounted. All stained sections were assessed on a light microscope (Nikon Eclipse SE EI R STG HNDL TRINOC-oil obj set with digital sight 1000 microscope camera).

### 2.9. Xenotumor Cell Transplant in NSG branded Mice

Resected tumors from untreated, GEM treated, OP+ASA and PUMP (OP+ASA) treated RAGxCγ PANC-1 xenograft mice were dissociated using collagenase and DNase. The xenotumor cells (5 × 10^5^) were co-cultured with HEK 293 cells in Matrigel to include a stromal cell component before injection into the mammary fat pads of NSG branded mice. The NSG mouse (NOD SCID gamma mouse) is a brand of immunodeficient laboratory mice developed and marketed by Jackson Laboratory (Bar Harbor, ME USA), which carries the strain NOD.Cg-Prkdcscid Il2rgtm1Wjl/SzJ. NSG branded mice lack mature T cells, B cells, and natural killer (NK) cells. NSG-branded mice are deficient in multiple cytokine signaling pathways and have many innate immunity defects. The referenced mice are described in detail at Jackson Laboratories. Mice were maintained in SPF isolators in the Animal Care Facility, Queen’s University, Kingston, Ontario K7L3N6, Canada. All mice were kept under sterile conditions in micro-isolators or air-filtered cages and were provided with autoclaved food and water. Mice used in the studies were approved by the Animal Care Committee, Queen’s University. Female mice between 6 and 8 weeks of age and an average weight of 30 g were used. Animals were sacrificed by cervical dislocation. Mouse weight and tumor volumes were recorded for 50 days, after which animals were sacrificed and necropsied.

### 2.10. Statistical Analysis

Data are presented as the means ± the standard error of the mean (SEM) from two repeats of each experiment, each performed in triplicate. All statistical analyses were performed with GraphPad Prism software. All results were compared with a one-way analysis of variance (ANOVA) and Fisher’s LSD test, with the following asterisks denoting statistical significance: * *p* ≤ 0.05, ** *p* ≤ 0.001, and *** *p* ≤ 0.0001.

## 3. Results

### 3.1. Critical Timing Protocol of Continuous Perfusion of ASA and OP via a Mini-Osmotic Pump Blocks the Tumor Growth and Metastases in Heterotopic Xenograft of Human Pancreatic PANC-1 Cancer Cells in RAG2xCγ Double Mutant Mice

To evaluate the efficacy of the continuous perfusion of ASA and OP, a mini-osmotic pump was surgically implanted subcutaneously into the flank of each animal following the development of a palpable tumor. Here, the preclinical in vivo anti-tumor activity of an ASA and OP-loaded pump (PUMP) was investigated in a RAG2xCγ double mutant xenograft mouse model of human pancreatic cancer previously reported by us [34]. The RAG2xCγ mice lack T cells, B cells, and NK cells. They are also deficient in cytokines involved in signaling, leading to better engraftment of human cells than any other published mouse strain. To determine the timing protocol of continuous perfusion of the PUMP (ASA + OP), half of the animals were sacked at the end of the pump perfusion period (42 days). The other half was monitored for an additional two weeks to evaluate any longer-term treatment effects (Figure 1A). Within the 42-day window of pump perfusion, the animals in the PUMP group, in combination with GEM, impeded the tumor growth rate compared to the untreated control (CTRL) and GEM-only treated animals (Figure 1B). For the animals maintained until the end of the study on day 121, the PUMP (ASA + OP) cohort maintained on GEM treatment had a 90% reduction in tumor volume compared to the CTRL and an 85% reduction in the GEM-only cohort (Figure 1B). This treatment protocol was also reflected in the wet tumor weight taken at necropsy, with minimal difference in tumor weight at the end of the pump timepoint (Figure 1C). However, at the study endpoint, there was a marked reduction in tumor weight in the PUMP (ASA + OP) treated animals compared to the CTRL and GEM animals (Figure 1D). These results indicate an effective, continuous tumor reduction after the pump emptied on day 42. The necropsy tumor images at the study endpoint showed more vascularized tumors for the CTRL and GEM cohorts than the PUMP (ASA + OP) cohort, suggesting that ASA and OP continuous treatment affected the angiogenesis or hypoxia tumor microenvironment (TME) (Figure 1E,F). The data in Figure 1G show the body condition weight to estimate a quantitative yet subjective method for evaluating the health of the animals following treatments.

Due to the highly metastatic nature of pancreatic cancer, we examined the liver and the lungs at necropsy, the two common sites of pancreatic metastases as previously reported by us [34]. Gross histology of the liver and lungs at day 98 (pump end point) and day 121 (study endpoint) necropsy were examined for metastatic nodules as well as H&E stained for the identification of micro-metastatic clusters (Figure 1). Interestingly, although there were not many metastatic nodules in the liver and lung at the pump endpoint, the GEM group had a visibly greater number of metastatic nodules at the study endpoint (Figure 1C, D). It is noteworthy that one of the pump cohort animals (mouse M4) had a large liver metastatic nodule (Figure 1D). Further examination of the tumor/pump necropsy in Figure 1F (M4) showed a bloody tissue lesion at the tumor site, which would contribute to the development of metastases. These data were confirmed following H&E staining for micro-metastases (liver and lung metastatic clusters within the tissue) (Figure 1E–H). The liver and lung weights were also recorded, indicating the metastatic load in the liver tissues (Figure 1P, Q) and lung tissues (Figure 1T,Y). Notably, the tumor volume of the GEM-treated cohort revealed an inflection of tumor growth at day 60 post-implantation. These data suggest that GEM chemotherapy induces chemoresistance in promoting tumor progression and growth (Figure 1B) and liver (Figure 1H–Q) and lung (Figure 1R–Y) metastatic spread of the disease, which were abrogated following continuous perfusion treatment with ASA + OP. Although there was not much difference at the pump endpoint, GEM-treated animals had a more significant number of micro-metastatic clusters in the liver and lungs than PUMP (ASA + OP)-treated animals at the study endpoint.

This initial study served as proof of concept for the continuous perfusion of ASA and OP via an osmotic pump delivery system. Collectively, these data suggest that a critical timing protocol of continuous perfusion of ASA and OP via an osmotic pump affects the tumor growth and metastases of pancreatic cancer.

### 3.2. Continuous Perfusion of ASA and OP Using an Osmotic Pump Compared to Weekly Injections in Combination with GEM Impedes Tumor Growth of Heterotopic Xenograft from Human Pancreatic PANC-1 Cancer Cells in RAG2xCγ Double Mutant Mice

Since we established that the continuous perfusion of ASA + OP in an osmotic pump presents a promising option for treating pancreatic cancer, we questioned whether ASA, OP, or ASA + OP loaded pumps would be effective treatment options compared to ASA + OP injections (INJ) for the treatment of pancreatic cancer. The rationale is that the combination of ASA and OP with GEM treatment significantly upends MiaPaCa-2 and PANC-1 pancreatic cancer cell viability, clonogenicity potential, and expression of critical extracellular matrix expression proteins, migration, and induces apoptosis, as previously reported by us [6].

In the next series of experiments, the study endpoint was selected to be ten days following the end of pump perfusion. Here, animals received a pump loaded with either ASA (50 mg/kg), OP (40 mg/kg), or ASA + OP (same dosages) or received 3× weekly injections (INJ) of ASA + OP (same dosages) together with GEM (100 mg/kg) once a week or were treated with GEM only (same weekly dose regimen) or remained untreated as control (CTRL) (Figure 2A).

At the study endpoint at day 93, the pump cohorts and INJ groups had a 95% reduced tumor volume compared to the CTRL and an 81% reduced tumor volume compared to the GEM-only group (Figure 2B). Due to the size of the tumors and lacerations, some of the animals in the CTRL cohort were sacked earlier and did not reach the study endpoint (Figure 2B,C). Additionally, one animal (M1) in the PUMP (ASA + OP) cohort had skin laceration at the pump site, and the pump was removed on day 60 (Figure 2C,K); skin suture was clipped and was continued in the study. Notably, the tumor from animal M1 continued to grow with weekly injections of GEM-indicated dosages. In addition, animal M4 in the PUMP (ASA + OP) group had a neck infection and laceration and was euthanized on day 56 (Figure 2C,K).

### 3.3. Continuous Perfusion of ASA and OP Abrogates GEM Chemoresistance of Heterotopic Xenografts of Human Pancreatic PANC-1 Cancer Cells in Developing Metastases in Liver and Lung of RAG2xCγ Double Mutant Mice

Similar to the results depicted in Figure 1, the continuous perfusion of pumps and INJ of ASA and OP in combination with GEM treatments significantly inhibited the metastatic burden on the liver (Figure 3A–D). As discussed previously, mouse M1 of the PUMP (ASA + OP) cohort lost the pump part way through the study, and this was reflected in the high metastatic burden seen in the liver (highlighted in circles) (Figure 3A). Excluding this mouse M1, animals receiving weekly ASA + OP injections (INJ), and all pump-treated animals had a significantly lower number of metastatic nodules in the liver than the untreated CTRL and the GEM-only animals (Figure 3B). Since most PDACs express cytokeratin [43], a pan-cytokeratin marker was used to identify PDAC metastatic tumor cells in the liver (Figure 3E). ASA/OP/GEM treatment protocols reduced positive staining for pan-cytokeratin compared to CTRL and GEM-only treated animals.

Not surprisingly, similar results were obtained upon examination of the lungs (Figure 3F–I). INJ and all pump groups combined with GEM had a significantly lower number of metastatic nodules in the lungs compared to the GEM-only group. Mouse M1 of the missing PUMP (ASA + OP) revealed 17 micro lung metastases (Figure 3I). These results demonstrated that GEM chemotherapy alone induces a metastatic program in the surviving tumor cells. In line with the data from the liver, stained lung tissue for pan-cytokeratin revealed fewer positive cells in the INJ and pump-treated cohort tumor tissues compared to the untreated CTRL and GEM-only counterparts (Figure 3J).

### 3.4. Continuous Perfusion of ASA and OP in Combination with GEM Inhibits Pancreatic Tumor Xenograft Cell Differentiation and Proliferation and Promotes Apoptosis in RAG2xCγ Double Mutant Mice

We have previously reported using the methylcellulose clonogenicity assay on MiaPaCa-2 and PANC-1 cells to determine metastatic, resistant pancreatic progenitor cells to proliferate and differentiate into colonies using in a semi-solid media [6]. PANC-1 cells revealed a statistically significant decreased in the clonogenicity potential following treatment with ASA + GEM, OP + GEM, and ASA + OP + GEM compared to the CTRL and GEM alone (*p* < 0.01) [6]. Here, the xenograft tumors at necropsy were fixed, paraffin-embedded, and stained with antibodies for pancreatic cell differentiation marker PDX-1, proliferation marker Ki67, and the TUNEL assay to measure apoptotic cells in the tumor (Figure 4).

PDX-1 marker is expressed during pancreatic development in the duct cells involved in islet cell differentiation but also has the potential to transform into several other cell types. For example, in pancreatic cancer, PDX-1 expression may de-differentiate tumor cells with more aggressive states [44]. The role of PDX-1 changes from tumor suppressive to oncogenic, with cells losing PDX-1 expression while undergoing EMT and PDX-1 loss being associated with poor patient outcomes [45]. Here, INJ and all pump group tumors had a significantly lower expression of PDX-1 compared to the CTRL and GEM-only tumors (Figure 4A,D).

Concerning the proliferation marker Ki67, it is noteworthy that the tumors from the GEM-only treated mice had a significantly higher number of Ki67-positive cells than all other treatment protocols (Figure 4B,E).

The TUNEL assay was performed on sections of paraffin-embedded tumor tissue to measure the expression of apoptotic cells. Here, tumors from INJ and PUMP (ASA + OP) treatments had a significantly higher number of positive cells, suggesting that the combination of ASA + OP with GEM promotes apoptosis of pancreatic tumor cells (Figure 4C,F). Tumors from the GEM-only treatment did result in a significantly high number of apoptotic cells, as expected, compared to the untreated CTRL (Figure 4C,F). It is noteworthy that all pump treatments significantly inhibited the anti-apoptotic BCL2 protein compared to the GEM cohort (Figure 4G). Not only the transcription factor c-myc is involved in enhancing cell proliferation, but it also regulates apoptosis. All treatment regimens significantly inhibited it compared to the GEM cohort (Figure 4H). Interestingly, the continuous perfusion of ASA and OP combined with GEM revealed a significant reduction of the proliferation marker, Ki67 (Figure 4E).

Normal processes in cellular growth, proliferation, metabolism, motility, survival, and apoptosis are regulated by the PI3K/Akt/mTOR signaling pathway. Aberrant PI3K/Akt activation may promote the survival and proliferation of tumor cells in many human cancers [46,47,48]. ASA has been reported to inhibit the PI3K/Akt/mTOR signaling pathway to mitigate GEM resistance and EMT in pancreatic cancer cells [49]. ASA was also reported to inhibit protein expression levels of PI3K, p-Akt significantly, and p-mTOR in PANC-1 cells compared to untreated controls [50]. However, these results have not been translated to preclinical animal studies. Here, we assessed the expression of mTOR, PI3K, and Akt1 on our tumor tissue sections following our treatment regimens (Figure 5).

Interestingly, all treatments decreased PI3K expression except for PUMP (OP) in combination with GEM (Figure 5A,D). To explain this later result, it is noteworthy that all treatment regimens had ASA except for the PUMP (OP) and GEM combination, which may be due to ASA’s unique inhibition of the PI3K/Akt/mTOR signaling pathway to mitigate GEM resistance [49]. Additionally, the significant increase in PI3K expression following PUMP (OP) and GEM treatment may be due to the specific targeting and activation of altered EGFRs by OP, as previously reported by us [34]. For Akt1, INJ and PUMP (ASA + OP) treatments had the highest Akt1 expression compared to the CTRL (Figure 5B,E). Due to the signaling cascade of the PI3K/Akt/mTOR pathway, we also analyzed mTOR expression. PUMP (OP) and PUMP (ASA + OP) and their combination with GEM significantly reduced the levels of mTOR expression compared to both the CTRL and GEM tumors (*p* < 0.0001) (Figure 5C,F).

In PDAC, the characteristic desmoplastic stroma results in a hypoxic core nutritional deprivation and ultimately promotes pancreatic cancer invasion and metastasis [51]. Hypoxia in PDAC has been reported to enhance apoptosis resistance induced by GEM via PI3K/Akt/NFκB pathways and partially through the MAPK signaling pathways [52]. There are also reports that ASA inhibits hypoxia-induced stemness in A549 lung cancer cells [53] and promotes EMT via HIF-1α-mediated transactivation of EMT-inducing factors [54]. Under hypoxic environment, tumor cells develop increased apoptotic resistance to chemotherapeutic drugs due to the overexpression of the anti-apoptotic Bcl2 family of proteins involved in the regulation of apoptotic cell death [55].

Here, we assessed whether ASA and OP combined with GEM would affect the expression levels of HIF-1α, HIF-2α, CAIX, and pimonidazole (Figure 6). Notably, hypoxia induces carbonic anhydrase IX (CAIX), which is functionally linked to acidosis, implicated in invasiveness, and correlated with therapeutic resistance [56]. Pimonidazole is a 2-nitroimidazole that forms stable adducts with thiol groups in proteins, peptides, and amino acids in hypoxic cells [57]. The expression of pimonidazole within tumors and cells is directly proportional to hypoxia levels.

The data depicted in Figure 6A,E show that PUMP(ASA) + GEM treatment decrease HIF-1α tumor expression. However, in contrast, there was an increased HIF-2α expression (Figure 6B,F) compared to the untreated CTRL and GEM-only treated tumors. In Figure 6C,G, CAIX tumor expression was upregulated in PUMP(ASA) + GEM compared to the CTRL and GEM tumors. At the same time, pimonidazole was significantly reduced (Figure 6D,H). Mice were injected intraperitoneally with 60  mg/kg pimonidazole hydrochloride from a stock solution containing 20  mg/ mL in sterile PBS for 1.5 h prior to euthanasia. It is noteworthy that CAIX is induced by hypoxia and is correlated with therapeutic resistance [56].

### 3.5. Continuous Perfusion of ASA and OP in Combination with GEM Alters the Cancer Stemness Potential and EMT Reprogramming of Pancreatic Tumors

Several studies have reported on the significant role of EMT in PDAC progression and chemoresistance, with key markers E-cadherin, N-cadherin, and EpCAM (ESA) [58,59,60]. EMT can induce the expression of CSC markers, suggesting an interplay between EMT plasticity and CSCs. EpCAM is primarily considered an adhesion molecule but has more recently been reported to regulate cell proliferation and cancer stemness [61]. EMT is also characterized by the loss of epithelial cell marker E-cadherin and the gain of mesenchymal marker N-cadherin. We questioned whether ASA and OP combined with GEM treatment alters EMT in tumor tissues. Here, ASA and OP combined with GEM treatment induced a significant increase in E-cadherin expression with a concomitant decrease in N-cadherin expression compared to both the untreated CTRL and the GEM-only treated tumors (Figure 7A,B,E,F). Furthermore, EpCAM and vimentin, two markers of EMT associated with a poor prognosis, decreased following ASA/OP/GEM treatment compared to the CTRL and GEM only tumors (Figure 7C,D,G,H).

Pancreatic cancer is associated with many CSCs markers that are negative prognostic factors and associated with tumor recurrence and clinical progressions, such as CD133, CD24, CD44, CXCR4, and ESA. Pancreatic CSCs resist GEM treatment and require alternative therapies [62]. CD36 has been reported to play a role in cancer and metastasis, involving tumor metabolism, immuno-editing, anti-angiogenic processes, and therapy resistance [63]. CD36 has also been reported to contribute to cancer progression and metastatic potential by activating CSCs, EMT, and chemoresistance [63]. CD44 + cells in slow-cycling human oral carcinomas express high levels of the fatty acid receptor CD36 and lipid metabolism genes to initiate metastasis [64]. CD44 has also been associated with GEM resistance and increased invasiveness. The clinical presence of CD36+ metastasis-initiating cells correlates with a poor prognosis, but CD36 inhibition impairs cellular metastasis, at least in human melanoma- and breast cancer-derived tumors [64].

Here, we show that ASA/OP/GEM treatment decreased CD36 and CD44 expression in tumors compared to the untreated CTRL and GEM-only tumors (Figure 7I,J,M,N). However, there was a non-significant change in the expression of CD24 (Figure 7K,O). Furthermore, PUMP (ASA) and INJ-treated tumors did have lower CXCR4 expression compared to the untreated CTRL and GEM-only tumors; however, this was lost in the PUMP(OP) and PUMP(ASA + OP) treated tumors (Figure 7L,P). Interestingly, Seeber et al. [65] reported on CXCR4 expression in patients with PDAC. Here, a subset of tumors with excellent responsiveness to immunotherapeutic approaches have high CXCR4 expression that is associated with improved survival and a pro-inflammatory phenotype. Don-Salu-Hewage et al. [66] reported that CXCR4 expression was found in the nuclear fractions using prostate cancer cell lines when compared to normal prostate epithelial cells. This nuclear CXCR4 pool used a nuclear transport pathway.

Notably, alcohol dehydrogenase-1 family member A1 (ALDH1A1) is a marker of chemoresistance and cancer stem-like properties in addition to CD44 and CD24. The ALDH1 enzyme is involved in the oxidation of retinol to retinoic acid and is an essential process for stem cells in their early differentiation [67]. The expression levels of CD44 and CD24 (CD44^+^/CD24^low^) are stem cell marker characteristics for enhanced invasion and metastases. For example, breast tumors expressing CD44^+^/CD24^low^ have been shown to exhibit enhanced invasion and metastasis [68,69]. As shown in Figure 7Q, the CD44/CD24 ratio expressed on INJ, PUMP (ASA), and PUMP (OP) treated tumor tissues was not significantly different from GEM-treated tumor tissue.

In contrast, the ratio of CD44/CD24 expressed on CTRL tumor tissue was significantly high compared to the GEM group. Interestingly, the ratio of CD44/CD24 expressed on the PUMP (ASA + OP) treated tumors was significantly reduced compared to the GEM group. These data support the evidence that untreated tumors have an invasive and metastatic cancer stem cell characteristic property. The PUMP (ASA + OP) treated tumors significantly reduced invasive and metastatic cancer stem cell characteristics. The ALDH1A1 expression on tumor treatment with INJ, PUMP (ASA), PUMP (OP) and PUMP (ASA + OP) in combination with GEM was significantly reduced compared to GEM treatment (Figure 7R). The ABCG2 biomarker that is recognized for the stemness phenotype was also significantly reduced in the tumor tissues following treatment with PUMP (OP) and PUMP (ASA + OP) (Figure 7S).

### 3.6. Tumorigenic and Metastatic Potential of Xenografts of PANC-1 Tumor Cells from Control, GEM, ASA + OP Injections and PUMP (ASA + OP) Treated Mice into the Mammary Fat Pads of NSG (NOD SCID Gamma) Branded Mice

To confirm the tumorigenic and metastatic potential of xenografts of PANC-1 tumor cells, the resected tumors from untreated CTRL, GEM treated, INJ and PUMP (ASA+OP) treated PANC-1 xenograft RAGxCγ mice were dissociated using collagenase and DNase. The xenotumor cells (5 × 10^5^) were co-cultured with HEK 293 cells in Matrigel. This protocol provides a stromal cell component before injection into the mammary fat pads of NSG (NOD SCID gamma) immunodeficient branded laboratory mice. The NSG-branded mice are immunodeficient in B cells, T cells, natural killer cells, innate immune and cytokine signaling pathways defects [70,71]. All animals were monitored for tumor volume every three days without any treatments. Figure 8A clearly shows that the GEM-treated xenotumors had the most significant tumorigenic potential euthanized at day 24 and macro- and micro-metastases in the lung and liver (Figure 8C–L).

The untreated CTRL xenotumors displayed the highest macro- and micro-metastases in the liver and lung tissues compared to the GEM and PUMP (ASA + OP) xenotumors. Notably, the tumorigenic potential of the xenotumors from the PUMP (ASA + OP) treatment had no tumor growth up to day 36, followed by a slight increase in tumor volume to the study endpoint on day 42. Additionally, the INJ and PUMP (ASA + OP) treated xenotumors had nearly negligible macro- and micro-metastases in the liver and lung compared to the CTRL and GEM xenotumors.

To confirm the long-term effects of the treatment protocols, the data depicted in Figure 8M,P indicated that the INJ and all pump cohort tumors had a significantly lower expression of PDX-1 compared to the CTRL and GEM-only tumors. The tumors from the GEM-only treated mice had a significantly higher number of Ki67 positive cells than all other treatment cohorts (Figure 8N,Q). Additionally, tumors from INJ and PUMP (ASA + OP) treatments had a significantly higher number of TUNEL-positive cells, suggesting that the combination of ASA + OP with GEM promotes apoptosis of pancreatic tumor cells (Figure 8O,R). Tumors from the GEM-only treatment did result in a significantly high number of apoptotic cells, as expected, compared to the untreated CTRL (Figure 8O,R). The xenotumors from the INJ and PUMP (ASA + OP) treatments exhibited continuous long-term effects for reducing PDX1 and Ki67 expressions. In contrast, there was a significant increase in the TUNEL apoptotic population of cells compared to the CTRL and GEM treated xenotumors (Figure 8O,R).

## 4. Discussion

In the present study, continuous parenteral perfusion of OP and ASA with gemcitabine treatment significantly disrupted tumor progression, and metastasis spread to the lung and liver. Additionally, the treatment protocol significantly upended the critical compensatory signaling markers such as PDX-1, Ki67, TUNEL, BCL-2, c-myc, and PI3K/Akt1/mTOR; hypoxia markers such as HIF-1α, HIF-2α, CAIX and pimonidazole; the EMT markers such as N- and E-cadherins, EpCAM, and vimentin; and CSC markers such as CD36, CD44, CD24, CXCR4, ALDH1, and ABCG2 in a mouse model of human pancreatic cancer. To confirm the tumorigenic and metastatic potential of xenografts of PANC-1 tumor cells, the resected tumors from untreated CTRL, GEM-treated, INJ and PUMP (ASA + OP)-treated PANC-1 xenograft were dissociated, and the xenotumor cells were injected into the mammary fat pads of NSG branded mice without treatment. ASA- and OP-treated xenotumors significantly inhibited the metastatic potential when transferred into animals. The INJ and PUMP (ASA + OP)-treated xenotumors had nearly negligible macro- and micro-metastases in the liver and lung compared to the CTRL and GEM xenotumors. The xenotumors from the INJ and PUMP (ASA + OP) treatments exhibited continuous long-term effects for reducing PDX1 and Ki67 expressions with a significant increase in the TUNEL apoptotic cell population compared to the CTRL and GEM-treated xenotumors.

One of the striking results we have noted in these experiments that may be particularly relevant to the clinical treatment of patients with pancreatic cancer is the ability of our experimental treatment protocol to control both tumor growth and metastatic progression for an extended period. We have seen a consistent pattern emerge with these experiments. Initially, the GEM chemotherapy-only treatment cohort compared to the chemotherapy plus experimental treatment cohorts parallel each other in the size of primary tumor growth. However, experimental treatment even at these stages is superior to chemotherapy alone. At approximately six weeks after tumor implantation, these tumor curves diverge suddenly, with the primary tumors in the chemotherapy-only group rapidly increasing in size while the primary tumors in the animals on the experimental treatments remain relatively stable. The premise is that this inflection point critically represents the transition between relative treatment effectiveness and treatment failure due to drug resistance.

The inflection point observed in our preclinical animal studies may be particularly relevant to the clinical treatment of patients with pancreatic cancer. Median progression-free survival in patients with metastatic pancreatic cancer is approximately six months with either 5FU/Irinotecan/Oxaliplatin or Abraxane with GEM [10,11]. Once disease progression is observed, the patient’s cancer becomes increasingly challenging to control. Death can quickly result if second-line therapies fail, as they often do. The rapid and dramatic increase in the size of the primary tumors in the chemotherapy-only group in our animals studied once this inflection point has been reached likely reflects what is observed in the clinic once patients become resistant to treatment. Additionally, the transition point where tumors are less responsive to therapy is essential and relevant to pancreatic cancer patients. Another issue to consider is the ‘stiffness’ of the pancreatic tissue after the therapy treatments. Several studies have indicated that this phenotypic tissue stiffness is correlated with therapy responsiveness and, unfortunately, resistance to therapy [72,73].

Overall, the experimental treatment using the continuous delivery of OP and ASA worked very well in controlling the primary tumor and minimizing metastases. The only exceptions in this work occurred when a significant infection developed in the animal or the pump had to be removed during the study. At the end of the study, tumor cells transplanted into NSG branded mice revealed that the animals treated with pump (ASA+OP) markedly had a much more extended period without growth than those treated with GEM alone. Because of the rapid growth of the GEM-only treated tumors, these animals had to be sacrificed much earlier than those receiving tumors from the pump-treated animals. Despite this significant difference in time of sacrifice between the experimental cohort and chemotherapy cohort, the animals that received the tumors from the experimental cohort had almost no metastatic disease. In contrast, chemotherapy only treated animals developed metastases rapidly.

We hypothesized that the effectiveness of our experimental treatment approach might be derived primarily from its ability to shut down compensatory EMT pathways triggered by the chemotherapy treatment itself. This compensatory EMT mechanism for developing drug resistance during cancer treatments has been empirically demonstrated in several different cancers already studied [74]. The ability of this experimental treatment to inhibit the EMT process was reflected in the increased expression of E-cadherin and downregulation of N-cadherin. Furthermore, stem cell enrichment observed in both the untreated and chemotherapy-only cohorts was significantly reduced with the experimental treatment compared with the chemotherapy-only cohort.

Based on the plausible hypothesis that drug resistance primarily derives from the CSC population, the ability to shut down EMT progression and stem cell enrichment during ongoing ASA and OP treatment with GEM chemotherapy is likely the reason for the effectiveness of our novel treatment strategy. During these experiments, the mice receiving the continuous infusion of OP and ASA in combination with GEM had control of tumor volume and limited metastases compared to GEM only group with a few exceptions. The three animals that failed to control either lost their pump or had a significant infection requiring early euthanasia. Mouse 4 in the first study (Figure 2) had a significant laceration at the site of pump implantation that may have interfered with the experimental treatment leading to a relative lack of treatment efficacy.

We used parenteral delivery of the OP to target Neu-1 and avoid the drug’s first metabolism to the carboxylate form ineffective targeting Neu-1 [40]. ASA was also employed based on our studies showing its ability to inhibit Neu-1 [6,7]. We have preliminary data on OP analogs on pancreatic cancer cells that certain analogs are an order of 10^5^ magnitudes more effective at inhibiting Neu-1 than the parental OP. However, we employed OP in our preclinical animal studies because it is an approved drug substance tested in healthy human subjects in phase 1 human clinical trial [75]. No significant adverse effects were observed in our animal studies and the human clinical trials, making regulatory approval to test this drug in a human clinical trial in patients with pancreatic cancer much more streamlined. It is noteworthy that the doses of ASA and OP used in our pump cohorts over the six-week study period were equivalent to approximately 2x the dosages of the ASA and the OP given by injection. These higher drug doses were employed to ensure a stable level of ASA and OP could be maintained throughout the entire study period. Notably, the injection of ASA and OP was also very effective at shutting down EMT and stem cell enrichment in this study. However, with few exceptions, the continuous infusion protocol appeared superior at controlling the disease.

One possible proviso objection to these studies and their potential translational impact is that we employed only single-agent GEM as the chemotherapy backbone. The standard of clinical care in the metastatic setting upfront is combination chemotherapy. Both GEM with Abraxane and combination 5-FU/oxaliplatin/irinotecan have shown superiority over single-agent gemcitabine in Phase III clinical trials regarding both response rate and median progression-free survival in pancreatic cancer [76]. In our opinion, this treatment does not detract from the potential translational relevance of our preclinical study. We predict that our experimental treatment protocol with the current clinical standard of combination chemotherapy would prove even more compelling given its ability to interfere with the induction of EMT likely induced by the chemotherapy treatment. We have reported that ASA and OP can sensitize pancreatic cancer cells to single-agent GEM chemotherapy [6]. This drug sensitization is predicted to be enhanced with a more effective chemotherapy regimen.

In summary, we present a novel therapeutic strategy targeting Neu-1 that can be employed with the current standard of care combination chemotherapy in patients newly diagnosed with metastatic pancreatic cancer. This therapeutic strategy is predicted to increase the response rate and median progression-free survival compared to the current standard of care in this very difficult to treat and often rapidly fatal malignancy.

## 5. Conclusions

Pancreatic cancer remains an enigmatic therapeutic challenge due to intrinsic chemoresistance and the rapid development of acquired chemoresistance during chemotherapy treatment. In the current study, a continuous infusion of OP with ASA in combination with GEM could disrupt the development of acquired chemoresistance and prevent rapid disease progression observed in the GEM-only cohort. When administered with the current standard chemotherapy treatment, this unique combinational and continuous perfusion therapy could represent a potential therapeutic breakthrough in this highly lethal and difficult to treat malignancy.

## 6. Patents

M.R.S. reports patents for the use of Neu1 sialidase inhibitors in cancer treatment (Canadian Pa-tent No. 2858,246; United States Patent No. US2015/0064282 A1; European Patent No. 11874886.2; Chinese Patent No. ZL201180076213.7; German Patent No. 602011064575.7; Italian Patent No. 502020000014650; UK Patent No. 2773340; Swedish Patent No. 2773340; Spanish Patent No. 2773340; Switzerland Patent No. 2773340; French Patent No. 2773340). M.R.S. reports a patent for the use of oseltamivir phosphate and analogs thereof to treat cancer (International PCT Patent No. PCT/CA2011/050690). W.W.H. and M.R.S. report a patent on a method to improve the effec-tiveness of anticancer therapies by exposing to an inflammatory stimulus prior to treatment (Canadian Patent No. PCT/CA2017/050765, pending). W.W.H. and M.R.S. report a patent on the compositions and methods for cancer treatment (Canadian Patent No. PCT/CA2017/050768, pending). W.W.H. and M.R.S. report that they are in the process of seeking approval from Health Canada to test the therapeutic treatment reported in this study in a human clinical trial.

## Figures and Tables

**Figure 1 cancers-14-03595-f001:**
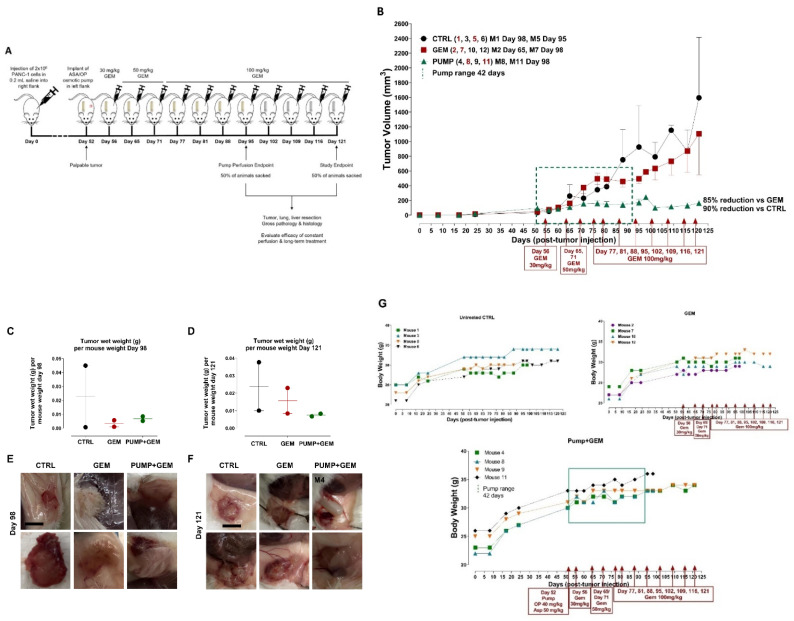
The study outline to evaluate the efficacy of constant perfusion of aspirin (ASA) and oseltamivir phosphate (OP) treatment at the end of pump perfusion (98 days post-pump implant) and the study endpoint (121 days post-pump implant). (**A**) Animals were divided into three cohorts: untreated control (CTRL), gemcitabine (GEM)-only treated, or implanted with an osmotic pump containing ASA (50 mg/kg) and OP (40 mg/kg) and treated weekly with GEM at the indicated dosages (**B**). Half of the animals in each cohort were euthanized at the end of the pump perfusion (day 98; mouse numbered in red) and the other half at the study endpoint (day 121, mouse numbered in black). Tumors, lungs, and livers were resected at necropsy and analyzed. (**B**) Measurement of tumor volume for each animal during the study. Pumps were implanted once each animal had a palpable tumor on day 52. The green square depicts the pump range. (**C**) Tumor wet weight at day 98 for each mouse and (**D**) Tumor wet weight at day 121 for each mouse. (**E**, **F**) Images of resected tumors at necropsy at the pump endpoints at days 98 and 121. (**G**) Body weight for untreated, GEM and PUMP (OP + ASA) days post tumor injection. (**H**) Quantification of metastatic nodules in the liver counted at the pump endpoint necropsy. (**I**) Gross histology of livers and metastatic nodules at necropsy at day 98. (**J**) Quantification of metastatic nodules counted at day 121 for the study endpoint necropsy. (**K**) Gross histology of livers and metastatic nodules at necropsy at day 121. (**L**) Quantifying the number of metastatic clusters counted from 2 + cuts of liver tissue at the end of the pump perfusion (day 98). (**M**) H&E staining of liver tissue at the pump endpoint. (**N**) Quantifying the number of metastatic clusters counted from 2 + cuts of liver tissue at the study endpoint (day 121). (**O**) H&E staining of liver tissue at the study endpoint. (**P**) Weight of livers at necropsy at day 98 and (**Q**) day 121. (**R**) Gross histology of lungs at necropsy at day 98, (**S**) Quantification of metastatic clusters in the lung from 2 + cuts of lung tissue at day 98, the pump endpoint necropsy, and (**T**) Lung weight at day 98. (**W**) Gross histology of lungs at necropsy at day 121. (**X**) Quantifying the number of metastatic clusters counted from 2 + cuts of lung tissue at the study endpoint day 121, (**Y**) Lung weight at day 121. H&E slides were visualized using the Nikon Eclipse EI R STG HNDL TRINOC microscope equipped with digital sight 1000 microscope camera with 20× objective. The scale bar represents 100 μm. Abbreviations: H&E—hematoxylin and eosin.

**Figure 2 cancers-14-03595-f002:**
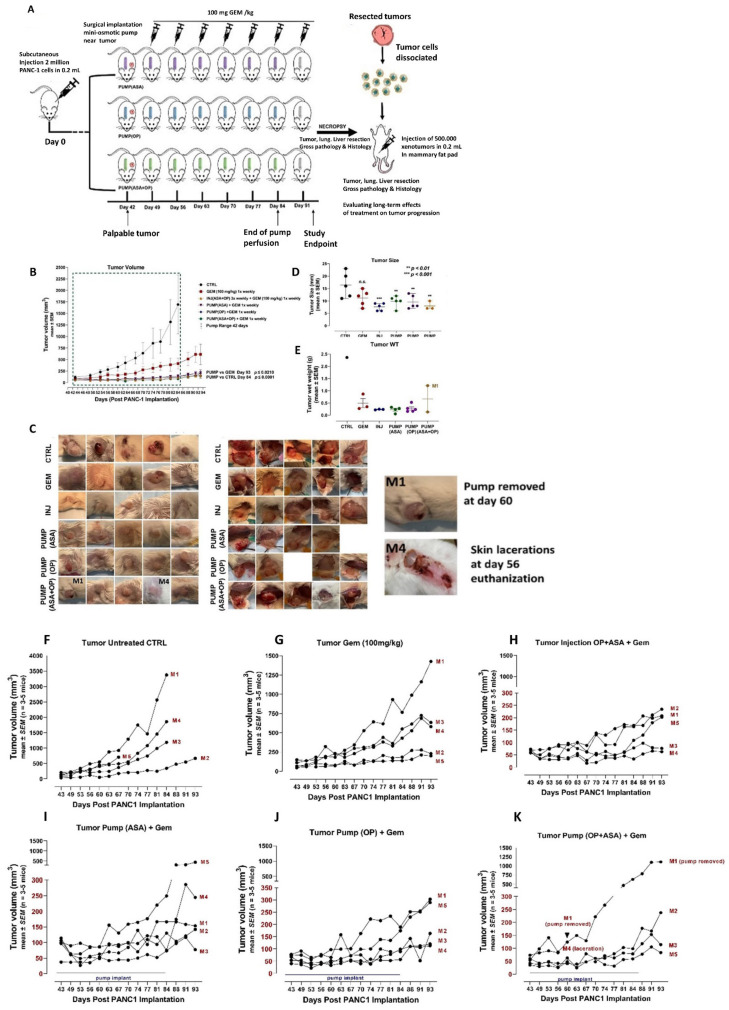
Evaluating the efficacy of constant perfusion of ASA, OP, and ASA + OP loaded pump compared to weekly injections combined with GEM in a mouse model of pancreatic cancer. (**A**) The study outline is illustrated to evaluate the efficacy of pump perfusion with ASA and OP compared to weekly IP injections of ASA + OP at the same dosages. The study was divided into two portions, one to evaluate treatment efficacy and the second to evaluate the long-term effects of treatment. Following implantation of 2 million pancreatic cancer PANC-1 cells subcutaneously, the treatment regimen began once a palpable 100 mm^3^ tumor formed. Animals were divided into six cohorts: untreated control (CTRL), GEM-only treated (100 mg/kg/wk), three weekly IP injections of ASA (50 mg/kg) + OP (40 mg/kg) plus GEM (100 mg/kg/wk), or surgically implanted with an osmotic pump loaded with either ASA, OP, or ASA + OP, at the same drug dosages. At the study endpoint, ten days after the end of pump perfusion, animals were sacked, and tumors, lungs, and livers were resected at necropsy and analyzed. Some tumor cells were dissociated from resected, and the xenotumor cells were injected into other animals. Once tumors reached their endpoint, animals were sacked, and tumors, lungs, and liver were resected for gross pathology and histology examination. (**B**) Measurement of tumor volume during the study. Pumps were surgically implanted on day 42 following PANC-1 cell implantation. The green square depicts the pump range. (**C**) images of necropsy tumors (**D**) Tumor length measured in mm, (**E**) wet weight at necropsy, and (**F**–**K**) tumor volume growth of individual animals. The one-way ANOVA Fisher test comparisons with 95% confidence, indicating asterisks for statistical significance compared to the untreated CTRL. Data are presented as mean ± SEM of individual animals. Abbreviations: ASA—aspirin; OP—oseltamivir phosphate; GEM—gemcitabine; CTRL—control; SEM—standard error of the mean.

**Figure 3 cancers-14-03595-f003:**
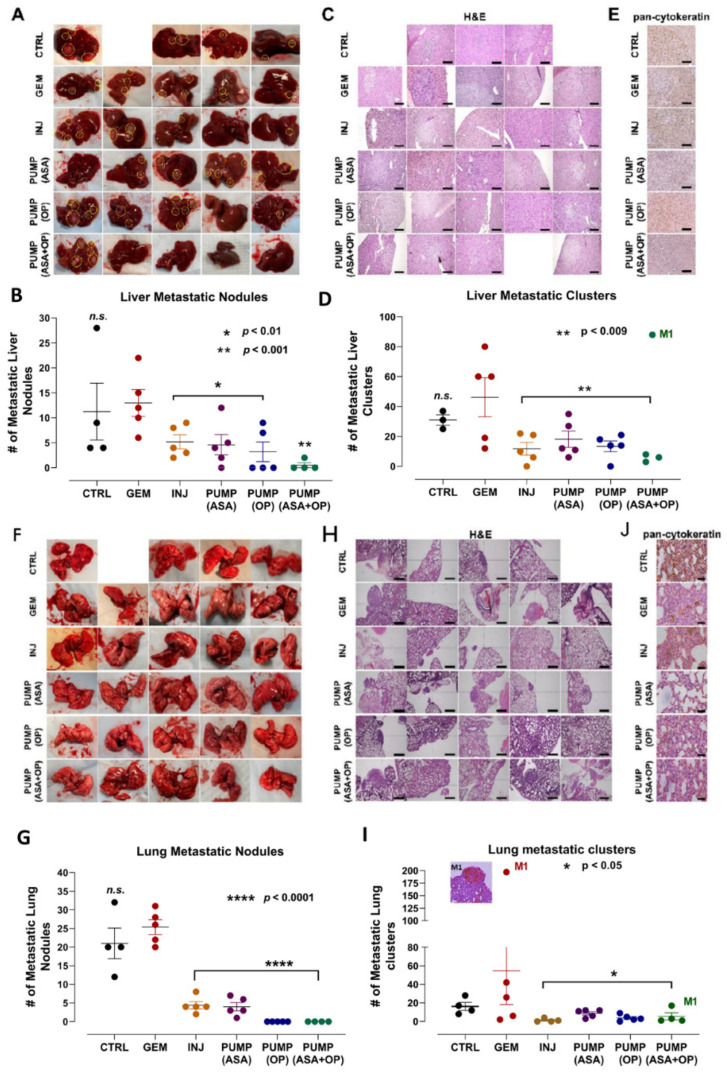
Liver and lung metastases following treatment with continuous perfusion of ASA + OP combined with GEM treatment of tumor xenografts of PANC-1 in RAG2xCγ double mutant mice. (**A**) Resected livers at necropsy. Yellow circles highlight visible metastatic nodules. (**B**) Quantification of the number of metastatic nodules counted. (**C**) Images of H&E staining of liver tissue were visualized using an epi-fluorescent microscope using the 10× objective. Scalebar represents 200 μm and (**D**) quantifying the number of metastatic clusters counted from multiple 5× cuts of liver tissue at necropsy. (**E**) Pan-cytokeratin staining of liver tissue was visualized with an epi-fluorescent microscope using the 20× objective. The scalebar represents 100 μm. (**F**) Resected lungs at necropsy with (**G**) quantification of the number of metastatic nodules counted. (**H**) Images of H&E staining of lung tissue were visualized using the Nikon Eclipse EI R STG HNDL TRINOC microscope equipped with digital sight 1000 microscope camera with the 10× objective. Scalebar represents 200 μm. (**I**) quantification of the number of metastatic clusters counted from multiple cuts of lung tissue at necropsy, and (**J**) Pan-cytokeratin staining of lung tissue visualized using an epi-fluorescent microscope using the 20× objective. The scalebar represents 100 μm. The one-way ANOVA Fisher test comparisons with 95% confidence, indicating asterisks for statistical significance compared to the untreated CTRL, unless otherwise shown. Data are presented as mean ± SEM of individual animals. Abbreviations: ASA—aspirin; OP—oseltamivir phosphate; GEM—gemcitabine; CTRL—control; SEM—standard error of the mean.

**Figure 4 cancers-14-03595-f004:**
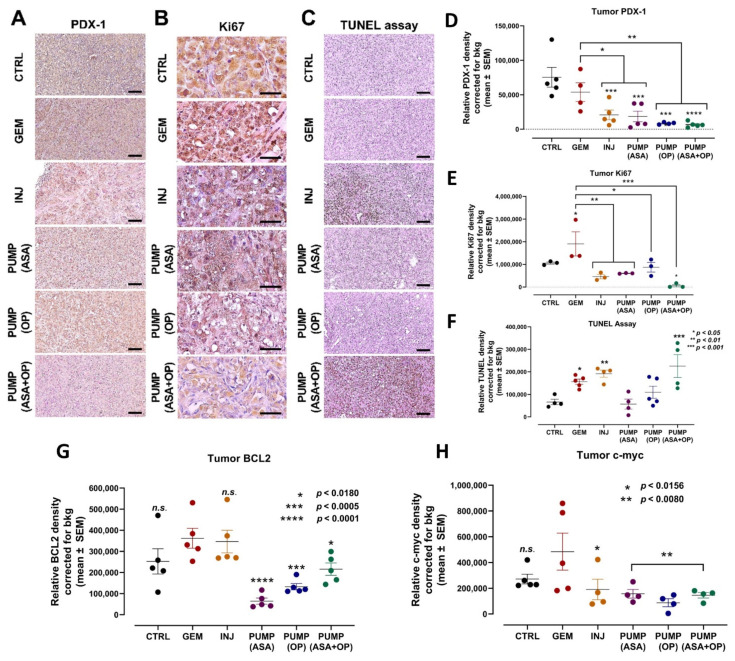
Continuous perfusion of ASA and OP in combination with GEM inhibits pancreatic tumor xenograft cell differentiation and proliferation and promotes apoptosis in RAG2xCγ double mutant mice. (**A**) Expression of pancreatic xenograft tumor cell differentiation marker PDX-1 following treatment with ASA/OP/GEM. Tissue images were captured using the Nikon Eclipse EI R STG HNDL TRINOC microscope equipped with digital sight 1000 microscope camera using the 20× objective. Scale bars represent 100 μm. (**B**) Expression of proliferation marker Ki67 following treatment with ASA/OP/GEM and visualized using an epi-fluorescent microscope using the 40× objective. Scale bars represent 200 μm. (**C**) TUNEL assay was performed on sections of paraffin-embedded tumor tissue and visualized using an epi-fluorescent microscope using the 20× objective. Scale bars represent 100 μm. Quantification of the relative density of (**D**) PDX-1, (**E**) Ki67, (**F**) TUNEL, as a measure of the number of positive cells, (**G**) BCL2 and (**H**) c-myc. Relative density was reported as mean ± SEM of 3–5 slides per treatment. The one-way ANOVA Fisher test comparisons with 95% confidence, indicating asterisks for statistical significance compared to the untreated CTRL, unless otherwise shown. Data are presented as mean ± SEM of individual staining. Abbreviations: ASA—aspirin; OP—oseltamivir phosphate; GEM—gemcitabine; CTRL—control; SEM—standard error of the mean.

**Figure 5 cancers-14-03595-f005:**
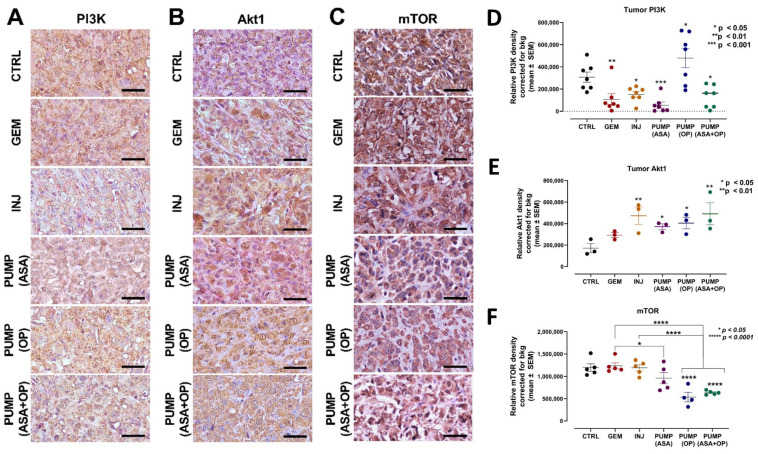
Continuous perfusion of ASA and OP in combination with GEM inhibits pancreatic tumor xenograft cell PI3K/Akt/mTOR signaling pathway in RAG2xCγ double mutant mice. (**A**) Expression of PI3K (**B**) Akt1, and (**C**) mTOR on pancreatic tumor tissue following treatment with ASA/OP/GEM. Tissue images were captured using the Nikon Eclipse EI R STG HNDL TRINOC microscope equipped with digital sight 1000 microscope camera using the 40× objective. Scale bars represent 200 μm. Quantification of the relative density of (**D**) PI3K, (**E**) Akt1, and (**F**) mTOR, as a measure of the number of positive cells. Relative density was reported as mean ± SEM of 3–5 slides per treatment. The one-way ANOVA Fisher test was used for comparison tests at 95% confidence intervals. The indicated asterisks represent the statistical significance compared to the untreated CTRL or otherwise shown. Abbreviations: GEM—gemcitabine; CTRL—control; OP—oseltamivir phosphate; ASA—aspirin; SEM—standard error of the mean.

**Figure 6 cancers-14-03595-f006:**
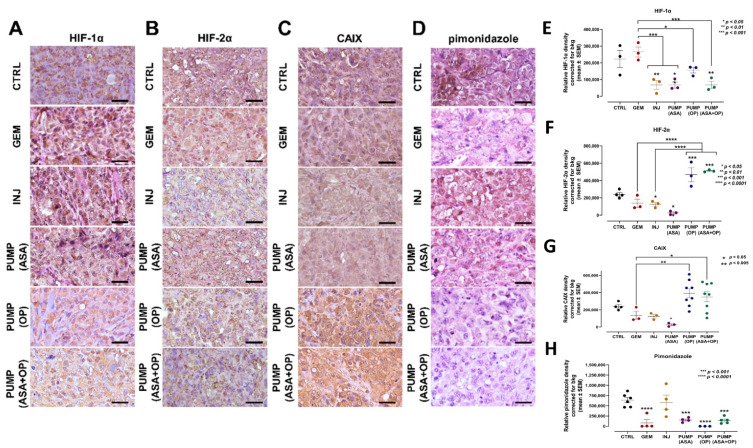
Aspirin, oseltamivir phosphate, and gemcitabine modify the expression of hypoxia markers in pancreatic cancer tumors. (**A**) Expression of HIF-1α (**B**) HIF-2α, (**C**) CAIX, and (**D**) pimonidazole in pancreatic tumor tissue following mice indicated treatments with ASA/OP in combination with GEM. For pimonidazole tumor tissues, mice received an IP injection of 15 mg of pimonidazole/mouse 1 h prior to euthanasia. Tissue images were captured using the Nikon Eclipse EI R STG HNDL TRINOC microscope equipped with digital sight 1000 microscope camera using the 40× objective. Scale bars represent 200 μm. Relative pixel density of (**E**) HIF-1α, (**F**) HIF-2α, (**G**) CAIX, and (**H**) pimonidazole. Relative density was reported as mean ± SEM of 3–5 slides per treatment. The one-way ANOVA Fisher test was used for comparison tests at 95% confidence intervals. The indicated asterisks represent the statistical significance compared to the untreated CTRL or otherwise shown. Abbreviations: CAIX—carbonic anhydrase 9; ASA—aspirin; OP—oseltamivir phosphate; GEM—gemcitabine; CTRL—control; SEM—standard error of the mean.

**Figure 7 cancers-14-03595-f007:**
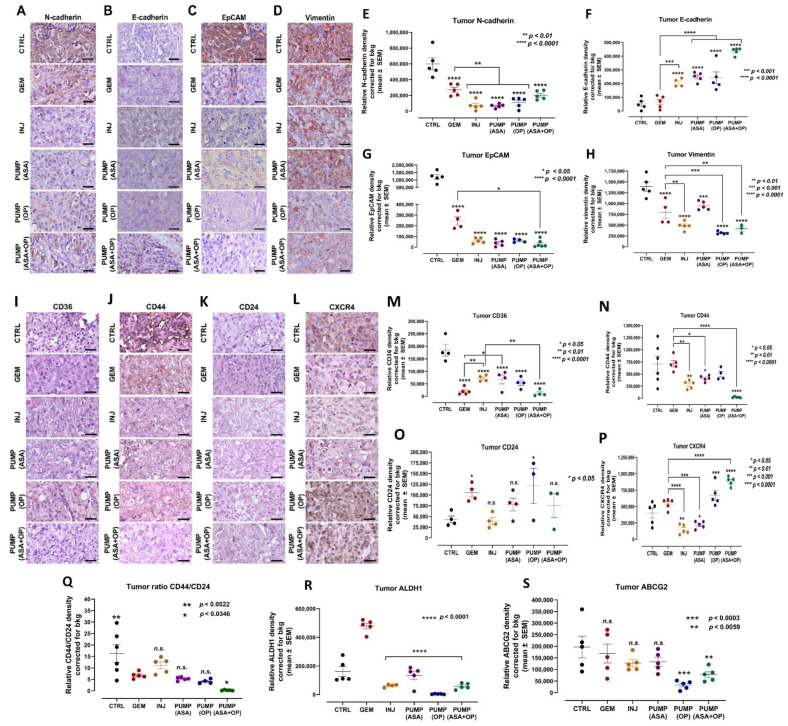
ASA and OP, in combination with GEM, inhibit the development of EMT markers in pancreatic tumors. (**A**) Expression of N-cadherin, (**B**) E-cadherin, (**C**) EpCAM, and (**D**) vimentin on pancreatic tumor tissue following treatment with ASA/OP/GEM. Tissue images were captured using the Nikon Eclipse EI R STG HNDL TRINOC microscope equipped with digital sight 1000 microscope camera using the 40× objective. Scale bars represent 200 μm. Relative density of (**E**) N-cadherin, (**F**) E-cadherin, (**G**) EpCAM, and (**H**) vimentin. Relative density was reported as mean ± SEM of 3–5 slides per treatment. (**I**–**S**) In combination with GEM, ASA and OP alter the stem cell population in pancreatic tumors. (**M**) Expression of CD36, (**N**) CD44, (**O**) CD24, (**P**) CXCR4, (**Q**) ratio CD44/24, (**R**) ALDH1 and (**S**) ABCG2 on pancreatic tumor tissue following treatment with ASA/OP/GEM. Tissue images were captured using the Nikon Eclipse EI R STG HNDL TRINOC microscope equipped with digital sight 1000 microscope camera using the 40× objective. Scale bars represent 200 μm. The one-way ANOVA Fisher test was used for comparison tests at 95% confidence intervals. The indicated asterisks represent the statistical significance compared to the untreated CTRL or otherwise shown. Abbreviations: ASA—aspirin; OP—oseltamivir phosphate; GEM—gemcitabine; CTRL—control; SEM—standard error of the mean.

**Figure 8 cancers-14-03595-f008:**
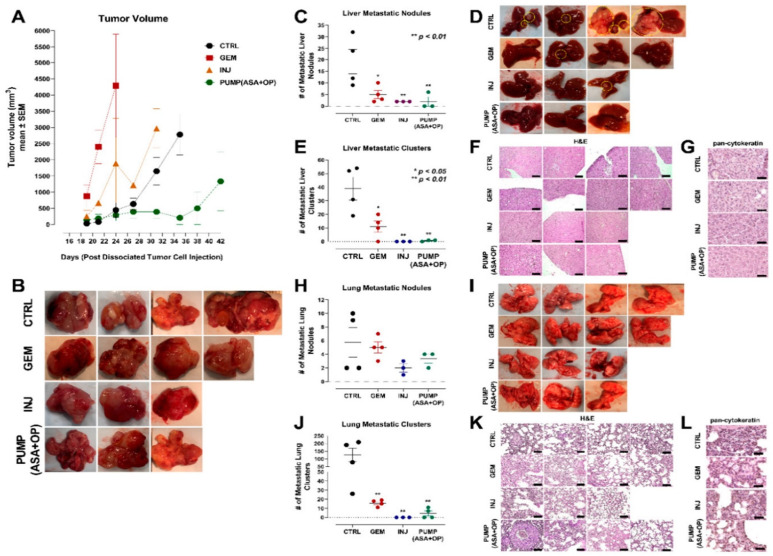
Tumorigenic and metastatic potential of xenografts of PANC-1 tumor cells from control, GEM, ASA + OP injections and PUMP (ASA + OP) treated mice into the mammary fat pads of NSG (NOD SCID gamma) branded mice. (**A**) Measurement of tumor volume during the study and (**B**) images of resected tumors at necropsy. (**C**) Quantification of the number of metastatic nodules counted in the liver and (**D**) Resected livers at necropsy. Yellow circles highlight visible metastatic nodules. (**E**) Quantifying the number of metastatic clusters counted from multiple cuts of liver tissue at necropsy. (**F**) Images of H&E staining of liver tissue were visualized using an epi-fluorescent microscope using the 10× objective. The scale bar represents 200 μm. (**G**) Pan-cytokeratin staining of liver tissue was visualized with an epi-fluorescent microscope using the 40× objective. (**H**) Quantification of the number of metastatic nodules counted in the lung and (**I**) Resected lungs at necropsy. Yellow circles highlight visible metastatic nodules. (**J**) Quantifying the number of metastatic clusters counted from multiple cuts of lung tissue at necropsy. (**K**) Images of H&E staining of lung tissue visualized using the Nikon Eclipse EI R STG HNDL TRINOC microscope equipped with digital sight 1000 microscope camera with 40× objective. 10× objective. The scale bar represents 200 μm. (**L**) Pan-cytokeratin staining of lung tissue was visualized with an epi-fluorescent microscope using the 40× objective. The scale bar represents 100 μm. (**M**–**R**) The long-term effect on pancreatic cancer cell differentiation PDX-1, proliferation Ki67, and TUNEL apoptosis of xenotumors of PANC-1 tumor cells from CTRL, GEM, ASA + OP INJ and PUMP (ASA + OP) treated mice into the mammary fat pads of NSG (NOD SCID gamma) branded mice. (**M**) Expression of pancreatic cell differentiation marker PDX-1. Tissue images were captured using the Nikon Eclipse EI R STG HNDL TRINOC microscope equipped with digital sight 1000 microscope camera using the 40× objective. Scale bars represent 200 μm. (**N**) The expression of proliferation marker Ki67 was visualized using an epi-fluorescent microscope using the 40× objective. Scale bars represent 200 μm. (**O**) TUNEL assay performed on sections of paraffin-embedded tumor tissue visualized using an epi-fluorescent microscope using the 40× objective. Scale bars represent 200 μm. Quantification of the relative density of (**P**) PDX-1, (**Q**) Ki67, and (**R**) TUNEL as a measure of the number of positive cells. Relative density was reported as mean ± SEM of 3 + slides per treatment. Asterisks denote significance compared to the untreated CTRL unless otherwise shown. Abbreviations: ASA—aspirin; OP—oseltamivir phosphate; GEM—gemcitabine; CTRL—control; SEM—standard error of the mean.

## Data Availability

All data needed to evaluate the conclusions in the paper are present in the paper. The preclinical data sets generated and analyzed during the current study are not publicly available but from the corresponding author upon reasonable request. The data will be provided following review and approval of a research proposal Statistical Analysis Plan and execution of a Data Sharing Agreement. The data will be accessible for 12 months for approved requests, with possible extensions considered. For more information on the process or to submit a request, contact szewczuk@queensu.ca.

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
