# Peer review of "Repositioning of Old Drugs for Novel Cancer Therapies: Continuous Therapeutic Perfusion of Aspirin and Oseltamivir Phosphate with Gemcitabine Treatment Disables Tumor Progression, Chemoresistance, and Metastases"

_cancers, 2022, doi:10.3390/cancers14153595_

Round 1

Reviewer 1 Report

In this study, Qorri et al. showed that repositioning the traditional cancer chemotherapies (use GEM as its chemotherapy backbone) to continuous perfusion of ASA and OP along with GEM treatment, via osmotic pumps implanted in mice. The authors provided substantial xenograft data together with statistical analysis to show the advantage of using a osmotic pump, containing ASA + OP and GEM, as a more effective pancreatic cancer treatment compared to traditional chemotherapy. The outcome is very promising, showing significant reduction fo tumor size compared to the control group and GEM-only group. The results are in general solid and have the potential to provide an improved therapeutic strategy to pancreatic cancer. Given the urgent need of developing effective treatments for pancreatic cancers, I support publication of this work, while I do have some questions and comments:

Major comments

1.     As shown in Figure 1A, why 30mg/kg of GEM is added 4 days after ASA/OP osmotic pump is implanted (reason of these 4 days interval)? What is the purpose of boosting up the dosage of GEM (from 30mg/kg, 50mg/kg to 100mg/kg)? Also, in Figure 1G, the labeling of days and treatments are intertwined with labeling on axis, made it hard to read and each graph is too small to look at and the numbers and legends are way too small.

2.     For the use of mice, the authors should have indicated how many mice in total they use and how many mice were in each experimental group. Although the authors gave some serial number for the mice (e.g. M4, M1) but the population group should still be indicated.

3.     I have some questions on the control experiment. Would it be better to add a group of experiment with mice only treated with PUMP (ASA+OP) but without GEM? Is it a concern that there is no control experiment with the solution (which contains for example DMSO) but not the drugs? How can one rule out the possibility that cell apoptosis was induced for example by DMSO. I do acknowledge that I am not an expert on the experimental design, so I expect the authors either reconsider the experimental design or justify their current one.

4.     The results that have shown in Figure 4 through Figure 7 mainly aim to show that the osmotic pump (ASA+OP with GEM) has the best efficacy in reducing tumor growth/proliferation by analyzing corresponding markers. However, the methods being used to see these various expression markers are almost the same (stained with different antibodies and take tissue images and perform TUNNEL assays). I wonder if some of these figures can be included in the supplemental instead so that the work will be more concise.

5.     At the discussion part, the authors mentioned that there were a total of three animals failed to have effective control due to lost of pumps or had serious infections when implanted with the pumps. The authors may want to provide some discussions on what would be some possible solutions to avoid infections or potential rejection reactions when performing the pump implantation.

Specific minor comments:

6.     Figure 1B: What are the numbers in the parenthesis (e.g., CTRL(1, 3, 5, 6), and what do the colors of the numbers mean? How many tumors (mice) are sampled for each curve? Do they count both primary and metastatic tumors? While I can find some information in the text, it might be clear to add the information in the caption as well.

7.     Figure 1C/D: What are the dots? Indicate variance? Is the color scheme same as in Fig. 1B? Each sampled from one mouse? Will it be more informative if one also plots each individual data point explicitly?

8.     Figure 1G: It might be helpful to adjust the font size, line width etc. the same as in panel B.

9.     What is the dose of ASP+OP injection? The same as the pump one?

10.  Figure 2: Shouldn’t the first panel be indexed? Also consider increasing the font size.

11.  Page 2, line 68: “EMT is a highly conserved developmental de-differentiation program whereby…” and in line 75: “EMT is a highly conserved developmental de-differentiation program whereby…” have the same meanings but the words being used are almost the same, could it be more concise?

12.  Page 2, line 61: “…patients often die rapidly from their disease”, “disease” should be plural. There are a few other places with minor grammar issues.

13.  Page 4, “Materials and Methods” section, in line 171 there is a big gap (blank space) bewteen “ASA” and “in”.

14.  Page 6 and Page 9, “Results” sections 3.1 and 3.2, line 274 and line 357, there are gaps after the subtitles but there are no gaps in other 3.x sections, please make sure the format is consistent.

Author Response

In this study, Qorri et al. showed that repositioning the traditional cancer chemotherapies (use GEM as its chemotherapy backbone) to continuous perfusion of ASA and OP along with GEM treatment, via osmotic pumps implanted in mice. The authors provided substantial xenograft data together with statistical analysis to show the advantage of using a osmotic pump, containing ASA + OP and GEM, as a more effective pancreatic cancer treatment compared to traditional chemotherapy. The outcome is very promising, showing significant reduction for tumor size compared to the control group and GEM-only group. The results are in general solid and have the potential to provide an improved therapeutic strategy to pancreatic cancer. Given the urgent need of developing effective treatments for pancreatic cancers, I support publication of this work, while I do have some questions and comments:

Major comments

  1. As shown in Figure 1A, why 30mg/kg of GEM is added 4 days after ASA/OP osmotic pump is implanted (reason of these 4 days interval)? What is the purpose of boosting up the dosage of GEM (from 30mg/kg, 50mg/kg to 100mg/kg)? Also, in Figure 1G, the labeling of days and treatments are intertwined with labeling on axis, made it hard to read and each graph is too small to look at and the numbers and legends are way too small.

Authors response: The reason why we added GEM 4 days after pump implantation is a mandated procedure from the Veterinarian so that animals would recover from the surgery. The purpose of boosting up the dosage of GEM was to monitor the animal health response as well as the tumor growth to the chemo drug following surgical implantation of the pump. We wanted to reach the 100 mg/kg dosage of GEM which would simulate the dosage used in clinical trials. In Fig1G, we have enlarged the image to see the numbers in the X-axis.

  1. For the use of mice, the authors should have indicated how many mice in total they use and how many mice were in each experimental group. Although the authors gave some serial number for the mice (e.g., M4, M1) but the population group should still be indicated.

Authors response: In Fig 1G, there were four (4) mice in each cohort of the experiment. In Fig2E, the tumor volume of each animal in the cohort study, days post PANC-1 implantation, is provided. In Figure 2, there were a total of five (5) animals in each of the six cohorts in the study design.

  1. I have some questions on the control experiment. Would it be better to add a group of experiment with mice only treated with PUMP (ASA+OP) but without GEM? Is it a concern that there is no control experiment with the solution (which contains for example DMSO) but not the drugs? How can one rule out the possibility that cell apoptosis was induced for example by DMSO. I do acknowledge that I am not an expert on the experimental design, so I expect the authors either reconsider the experimental design or justify their current one.

Authors response: Thank you for this comment regarding the PUMP (ASA+OP) without GEM group. It is noteworthy that the design of the present experimental study is to test the therapeutic treatment reported here and to simulate it in a human clinical trial of patients with aggressive pancreatic cancer who will be under standard clinical chemotherapeutic gemcitabine as mandated with approval by Health Canada Clinical Trials scientific and clinical review.

       The other issue is that the Animal Care Committee requests reduction in the use of animals in the experimental design. To this end, we have reported that the treatment of PANC-1 and MiaPaCa-2 pancreatic cancer cells with ASA and OP without GEM was significantly effective in reducing cell viability, invasion, clonogenicity, migration, as well as Caspase 3/7 Apoptosis Assay, tube formation assay and analysis, and WST-1 cell viability assay, MCTS spheroids, morphologic changes, cell viability, apoptosis activity and the expression levels of ALDH1A1, CD44 and CD24 on the cell surface, MDA-MB231 triple-negative breast cancer (Qorri, Harless, Szewczuk. 2020 Drug Des Devel Ther. 2020 144149-4167; Qorri, Mokhtari, et. Cancers 2022, 4, 1374; and Sambi, Samuel et al. (2020) Drug Design, Development and Therapy 14 1995–2019).

       With the comment to use MiaPaCa-2 cells for in vivo studies, we have reported the proof-of-evidence for a therapeutic targeting of Neu1 with OP without chemotherapeutic drug in impeding human pancreatic cancer growth and metastatic spread in heterotopic xenografts of eGFP-MiaPaCa-2 tumors growing in RAGxCγ double mutant mice (Gilmour et al. 2013 Cellular Signalling 25-2587). Here, OP therapy at 100 mg/kg daily dosage intraperitoneally expectedly impeded human pancreatic tumor growth in a time-to-progression growth rate compared to the untreated cohort. Following OP treatment, there was no significant increase in the time-to-progression tumor growth rate compared to a significant tumor growth rate for the untreated cohort. To confirm these results, we also found at necropsy that there was a significant reduction in tumor size and tumor weights at day 47 post-implantation taken from the OP treated tumor-bearing mice compared to the untreated cohort. This dose regime of OP had no side effects as determined by body weight and body condition scoring. Also, OP-treated cohort exhibited a reduction of phosphorylation of EGFR-Tyr1173, Stat1-Tyr701, Akt-Thr308, PDGFRα-Tyr754 and NFκBp65-Ser311 but an increase in phospho-Smad2-Ser465/467 and -VEGFR2-Tyr1175 in the tumor lysates from the xenografts of human eGFP-MiaPaCa-2 tumor-bearing mice. The findings identified a novel promising alternate therapeutic treatment of human pancreatic cancer.

  1. The results that have shown in Figure 4 through Figure 7 mainly aim to show that the osmotic pump (ASA+OP with GEM) has the best efficacy in reducing tumor growth/proliferation by analyzing corresponding markers. However, the methods being used to see these various expression markers are almost the same (stained with different antibodies and take tissue images and perform TUNNEL assays). I wonder if some of these figures can be included in the supplemental instead so that the work will be more concise.

Authors response: We believe that all of the data presented in the study is important to establish the proof-of-evidence for the efficacy of continuous therapeutic parenteral perfusion of OP and ASA with gemcitabine (GEM) treatment to significantly disrupt tumor progression, critical compensatory signaling mechanisms, EMT program, CSC, and metastases in a pre-clinical mouse model of human pancreatic cancer.

  1. At the discussion part, the authors mentioned that there was a total of three animals failed to have effective control due to lost of pumps or had serious infections when implanted with the pumps. The authors may want to provide some discussions on what would be some possible solutions to avoid infections or potential rejection reactions when performing the pump implantation.

 Authors response: It is important to note that one animal whose pump (ASA+OP) was removed at day 60 due to skin pump protrusion (Fig2B) was continued to the end point of the study after pump removal and wound sutured. These data provided an important proof-of-evidence that the continuous perfusion of ASA and OP with GEM treatment shows efficacy of the treatment protocol. The data provided an important internal negative control in the study cohort. Mouse M4 was euthanized due to skin infections. In clinical trials with patients with aggressive pancreatic cancer, the delivery of the continuous perfusion of OP plus ASA will be using IV pump perfusion and monitored.

Specific minor comments:

  1. Figure 1B: What are the numbers in the parenthesis (e.g., CTRL(1, 3, 5, 6), and what do the colors of the numbers mean? How many tumors (mice) are sampled for each curve? Do they count both primary and metastatic tumors? While I can find some information in the text, it might be clear to add the information in the caption as well.

Authors response: The numbers in parenthesis in Fig1B are the mouse numbered in the experiment. We have added in the figure legend, “(B). Half of the animals in each cohort were sacked at the end of the pump perfusion (day 98; mouse numbered in red) and the other half at the study endpoint (day 121, mouse numbered in black).” The number of mice sampled for each cohort curve is four (4) as indicated in parenthesis and legend. They are both primary and metastatic tumors.

  1. Figure 1C/D: What are the dots? Indicate variance? Is the color scheme same as in Fig. 1B? Each sampled from one mouse? Will it be more informative if one also plots each individual data point explicitly?

Authors response: the dots in Fig.1C/D represent tumor weight wet for each mouse. The figure legend for Fig1 C/D is revised as “(C) Tumor wet weight at day 98 for each mouse and (D) Tumor wet weight at day 121 for each mouse.” Each individual data point is explicit in the Fig.1C/D.

  1. Figure 1G: It might be helpful to adjust the font size, line width etc. the same as in panel B.

DONE

  1. What is the dose of ASP+OP injection? The same as the pump one?

YES

  1. Figure 2: Shouldn’t the first panel be indexed? Also consider increasing the font size.

The image has been enlarged. The legend indicates the study outline.

  1. Page 2, line 68: “EMT is a highly conserved developmental de-differentiation program whereby…” and in line 75: “EMT is a highly conserved developmental de-differentiation program whereby…” have the same meanings but the words being used are almost the same, could it be more concise?

Thank you for this comment. We have changed the last EMT issue to, “During the EMT de-differentiation program, more differentiated epithelial cells can acquire properties of stem cells.”

  1. Page 2, line 61: “…patients often die rapidly from their disease”, “disease” should be plural. There are a few other places with minor grammar issues.

Done

  1. Page 4, “Materials and Methods” section, in line 171 there is a big gap (blank space) bewteen “ASA” and “in”

DONE

  1. Page 6 and Page 9, “Results” sections 3.1 and 3.2, line 274 and line 357, there are gaps after the subtitles but there are no gaps in other 3.x sections, please make sure the format is consistent.

DONE

Reviewer 2 Report

Review

Repositioning of Old Drugs for Novel Cancer Therapies: Continuous Therapeutic Perfusion of Aspirin and Oseltamivir Phosphate with Gemcitabine Treatment Disables Tumor Progression, Chemoresistance, EMT program, Cancer Stem Cells, and Metastases

-------------------------------------------------------------------------------------------------------------------

The title is too long, please shorten the title for clarity (example below).

Repositioning of Old Drugs for Novel Cancer Therapies: Continuous Perfusion of Aspirin and Oseltamivir Phosphate with Gemcitabine Disables Pancreatic Tumor Metastasis and Chemoresistance

The manuscript is a well-designed study focusing on targeting pancreatic cancer chemotherapy resistance to Gemcitibine (Gem). There are apparent mechanisms that pancreatic cells intrinsically use to evade chemotherapy toxicity, while upregulating EGFR, Neu-1, VEGF, PDGFR, and other signaling factors. Another interesting component of the manuscript was to study shifts in epithelial-to-mesenchymal transition (EMT) in the PANC-1 and PANC-1-GemR cells to determine how the phenotypes of the cells can contribute to chemoresistance over an extended period. Osmotic pump administration of oseltamivir phosphate (OP) and aspirin (ASA) to the mouse models treated based on a clinical schedule with Gem represents a relevant chemotherapy study. Gem treatment is clinically relevant because it assists in understanding how patients develop these responses and/or subsequent resistance to chemotherapy.

Line 54: Please add references to describe if the authors demonstrate the “intrinsic resistance of pancreatic cancer” in their laboratory. If so, please add references. Is the work in vitro, in vivo, or clinical?

Line 59: Is “shrinkage” quantitated by volume, weight, or size of the tumors? How is “shrinkage” measured?

Line 71: Please replace “resistance to apoptosis” with “resist apoptosis”

Line 75: Please replace “micro metastasis” with “micro-metastasis”

Line 75: Reword this sentence because it repeats from line 68

Lines 73-78 Due to highly metastatic properties of pancreatic cancer, please expand on descriptions of 1) metastasis formation genes with current manuscript, 2) EMT-regulated genes that promote and block chemotherapy response/resistance

Line 80: What is the drug described? This is a critical statement related to EMT processes in this paper.

Line 84: Which cells are used? Are they cell lines? (breast, pancreatic)

Line 87: Please state what ASA (aspirin) is again. I had to scroll up to the Abstract to find out what ASA meant. This is just to make it easier for the reader.

Line 88: What cancer? Please explain what “live” means here, it is not a widely use technique. The live monitoring of the osmotic pumps in mice is an essential part of the project so it is necessary to describe it briefly here.

Line 90: Are MiaPaCa-2 the pancreatic cancer control or baseline cell line here?

Line 94: To prevent a run on sentence, please end the sentence at “[24-27].” Start a new sentence perhaps by stating, “All of these signaling complexes are upregulated and….”.

Line 96: What receptors are you describing? EGFR?

Line 102: Please add a comma after “pathway, but”.

Line 131: Please add information about MiaPaCa-2 cells.

Line 142: What percentage of saline was used?

Line 143: Was the PBS used a 1x concentration?

Line 144: What percentage of saline was used?

Line 151: What is SPF? Please explain here.

Line 161: Please replace “PANC1” with “PANC-1”.

Line 163: What percentage of saline was used?

Figure 1A: Please add metrics or reference to Methods of “palpable tumor” on the figure (i.e., palpable tumor was measured at 100 mm3). Please change “sacked” to “sacrificed” or “euthanized”

Figure 1B: Why was Gem increased from 30, 50 to 100 mg/kg? What is the rationale for treating the mice on certain days (55, 65, 77, etc.)?

Figure 2: The data in this figure appears to indicate that the OP+ASA+Gem combination therapy was more effective at reducing tumor volumes over time (based on Y-axis values). There is apparent variability between each individual mouse, but I understand that based on the representative images shown and the quantitation that combination is a better option. The osmotic pump with the combination therapy is even more effective.

Figure 4: The data presented is very convincing. I didn’t know of a pancreatic cancer marker named PDX-1 until now and this makes the study more interesting. The acronym PDX also made me think if the authors would consider using PDX from pancreatic cancer patients and the same therapeutic conditions in this study? This should not be added to this study, but in the future this would provide even more clinical relevance and justify the use of the pump with standard of care.

Figure 5: I appreciate the authors’ inclusion of the PI3K/Akt/mTOR signaling because of the long-standing literature on pancreatic cancer regulation of the processes listed in Lines 484-487. A very important publication to consider adding to this paper is cited below.

Lee S-W, Zhang Y, Jung M, Cruz N, Alas B, Commisso C. 2019. EGFR-Pak signaling selectively regulates glutamine deprivation-induced macropinocytosis. Developmental Cell. 50:381-392.

Since most figures focused on the tumor microenvironment, it is also important to recognize potential influence from the surrounding stroma of the pancreas in future studies. The influence of the stroma may regulate or even block therapeutic efficacy in the cells used to test the pump with OP, ASA, and Gem.

Also, is there a way to analyze Neu-1 by Western blot or at transcriptional levels? I have not seen its expression here, or is it assessed by activity level? Or is Neu-1 assessed indirectly by other proteins/genes? I am not familiar with Neu-1 but I am wondering if the authors can show it’s activity or expression changing in the figures because I don’t see it yet.

Please improve quality of the images on the box and whiskers plots. When I zoom in to higher than 200%, the quality of the images goes down and makes it hard to read the data. Some of the figures have artifacts on the graphs that must be removed to improve the viewing quality of the data. Please improve the quality of the IHC images, it is hard to see the staining at higher than 100% zoom. Also consider adding arrows to the IHC images where you have positive staining if it difficult to see.

Figure 6: Data is very convincing, no changes needed except for those listed to improve the image quality. Statistics for all figures are sound and relevant so far.

Figure 7: Most of the markers for EMT and possible MET decreased with the pump and combination therapy, but how do the authors plan to address the significant increase in CXCR4 expression in vivo? CXCR4 is a well-known marker that is induced by SDF1alpha in a nuclear-exclusive manner (citation below). So, perhaps the CXCR4 staining is mostly nuclear? Or maybe there is a way to only stain the non-nuclear form of this protein to gain a better understanding. This marker may bring more criticism to this figure than it is worth presenting based on its variable regulatory patterns even in other cancer cells.

Don-Salu-Hewage AS, Chan SY, McAndrews KM, Chetram MA, Dawson MR, Bethea DA, Hinton CV. 2013. Cysteine (C)-x-C receptor 4 undergoes transportin 1-depedent nuclear localization and remains functional at the nucleus of metastatic prostate cancer cells. PLoS One. 8(2):e57194.

CD24 inhibition is too variable to be convincing. Please increase the sample size to make sure that the expression is truly increased.

Discussion: Please consider reiterating the significance of the study with a brief summary. Please answer questions in a brief paragraph about why pancreatic cancer metastasis is a problem, why the current standard of care chemotherapy has lower efficacy in these patients, and how you addressed it. After reading through a large amount of data, the reader needs to be reacquainted with the scope of what you investigated. The first two paragraphs are out of order. Please discuss the data in the order that it is presented in the Results section to prevent confusing the reader. The data also would be more convincing if the authors repeated these experiments in the future with the seemingly more aggressive MiaPaCa-2 cells. I didn’t see data in this manuscript for these cells and I don’t see supplemental figures. In the authors’ previous work they showed that the triple combination therapy similarly regulated migration in both MiaPaCa-2 and PANC-1. Gem was more effective at reducing migration in PANC-1 compared to MiaPaCa-2 cells, but it may be beneficial to see if MiaPaCa-2 can be used as a tool to study the ‘inflection’ period that the authors discussed. The transition point where tumors are less responsive to therapy is important when presenting work that is clearly relevant to pancreatic cancer patients. Another point to look into is the ‘stiffness’ of the pancreatic tissue after the treatments. Several studies have indicated that this phenotype is correlated with therapy responsiveness and unfortunately resistant to therapy (citations below).

Nguyen AV, Nyberg KD, et al. 2018. Stiffness of pancreatic cancer cells is associated with increased invasive potential. Integr Biol (Camb). 8(12):1232-1245.

Kai F, Laklai H, Weaver V. Trends Cell Biol. 2016;26:1–12.

Author Response

Repositioning of Old Drugs for Novel Cancer Therapies: Continuous Therapeutic Perfusion of Aspirin and Oseltamivir Phosphate with Gemcitabine Treatment Disables Tumor Progression, Chemoresistance, EMT program, Cancer Stem Cells, and Metastases

-------------------------------------------------------------------------------------------------------------------

The title is too long, please shorten the title for clarity (example below).

Repositioning of Old Drugs for Novel Cancer Therapies: Continuous Perfusion of Aspirin and Oseltamivir Phosphate with Gemcitabine Disables Pancreatic Tumor Metastasis and Chemoresistance

Authors response: Thank you for your comment on the title. We would like to change it to “Repositioning of Old Drugs for Novel Cancer Therapies: Continuous Perfusion of Aspirin and Oseltamivir Phosphate with Gemcitabine Disables Pancreatic Tumor Progression, Chemoresistance and Metastasis”

The manuscript is a well-designed study focusing on targeting pancreatic cancer chemotherapy resistance to Gemcitabine (Gem). There are apparent mechanisms that pancreatic cells intrinsically use to evade chemotherapy toxicity, while upregulating EGFR, Neu-1, VEGF, PDGFR, and other signaling factors. Another interesting component of the manuscript was to study shifts in epithelial-to-mesenchymal transition (EMT) in the PANC-1 and PANC-1-GemR cells to determine how the phenotypes of the cells can contribute to chemoresistance over an extended period. Osmotic pump administration of oseltamivir phosphate (OP) and aspirin (ASA) to the mouse models treated based on a clinical schedule with Gem represents a relevant chemotherapy study. Gem treatment is clinically relevant because it assists in understanding how patients develop these responses and/or subsequent resistance to chemotherapy.

Authors response: Thank you for your comment. With regard to study shifts in epithelial-to-mesenchymal transition (EMT) in the PANC-1 and PANC-1-GemR cells and to determine how the phenotypes of the cells can contribute to chemoresistance, we have reported on this issue (Qorri, Harless, Szewczuk. 2020 Drug Des Devel Ther. 2020 144149-4167).

Line 54: Please add references to describe if the authors demonstrate the “intrinsic resistance of pancreatic cancer” in their laboratory. If so, please add references. Is the work in vitro, in vivo, or clinical?

Authors response: Thank you for this comment. We have added the appropriate references for work in the laboratory on intrinsic resistance of pancreatic cancer cells.”Intrinsic resistance of pancreatic cancer cells to chemotherapy treatment is well recognized clinically and, in the laboratory [5-9]. Line 52. The work was done in vitro.

Line 59: Is “shrinkage” quantitated by volume, weight, or size of the tumors? How is “shrinkage” measured?

Authors response: “An estimated 20-30% of patients will experience a response with these treatments manifested by shrinkage of the primary tumor and metastases.”Line 59.  Clinically, 50 % reduction in the size of a cancerous mass may reduce symptoms strikingly and produce impressive changes on physical or radiographic examination. It generally represents a relatively small reduction in the body's tumor burden, i.e., the total number of cancer cells present in the body.

Line 71: Please replace “resistance to apoptosis” with “resist apoptosis”

Line 75: Please replace “micro metastasis” with “micro-metastasis”

Line 75: Reword this sentence because it repeats from line 68

Authors response: DONE

Lines 73-78 Due to highly metastatic properties of pancreatic cancer, please expand on descriptions of 1) metastasis formation genes with current manuscript, 2) EMT-regulated genes that promote and block chemotherapy response/resistance

Authors response: DONE

Line 80: What is the drug described? This is a critical statement related to EMT processes in this paper.

Authors response: Thank you for this comment. We have included the following text, “For example, cancer cells that have undergone the EMT process, even a partial EMT, can see an increase in the half-maximal inhibitory concentration (IC50) dose of a chemotherapy drug gemcitabine by 10-fold. For example, O’Shea et al. [22] reported that oseltamivir phosphate (OP) upended the chemoresistance of PANC-1 to cisplatin and gemcitabine alone or in their combination in a dose-dependent manner. Also, OP reversed the EMT characteristic of N-cadherin to E-cadherin changes associated with resistance to cisplatin and gemcitabine drug therapy. It is noteworthy that the epidermal growth factor receptor (EGFR) is critical in inducing the EMT program in pancreatic cancer [23].”

Line 84: Which cells are used? Are they cell lines? (breast, pancreatic)

Authors response: human triple-negative MDA-MB-231 breast cells [33] and pancreatic MiaPaCa-2 [34] cancer cell lines  

Line 87: Please state what ASA (aspirin) is again. I had to scroll up to the Abstract to find out what ASA meant. This is just to make it easier for the reader.

Authors response: DONE

Line 88: What cancer? Please explain what “live” means here, it is not a widely use technique. The live monitoring of the osmotic pumps in mice is an essential part of the project so it is necessary to describe it briefly here.

Authors response: Removed “live” in line 88

Line 90: Are MiaPaCa-2 the pancreatic cancer control or baseline cell line here?

Authors response: MiaPaCa-2 (polymorphism) expresses CK5.6, AE1/AE3, E-cadherin, vimentin, chromogranin A, synaptophysin, SSTR2 and NTR1 but not CD56. PANC-1 (pleomorphism) expresses CK56, MNF-116, vimentin, chromogranin A, CD56 and SSTR2 but not E-cadherin, synaptophysin or NTR1. MIA PaCA-1 is CD24−, CD44+/++, CD326−/+ and CD133/1−, while PANC-1 is CD24−/+, CD44+, CD326−/+ and CD133/1−. Both cell lines have KRAS and TP53 mutations. So, MiaPaCa-2 cell line was used as a baseline in those studies.

Line 94: To prevent a run on sentence, please end the sentence at “[24-27].” Start a new sentence perhaps by stating, “All of these signaling complexes are upregulated and….”.

Authors response: DONE

Line 96: What receptors are you describing? EGFR?

Authors response: “Ligand binding to its receptor” refers to all of those mentioned in the previous sentence - including the epidermal growth factor receptor (EGFR) [19], insulin receptor (IR) [22], and the nerve growth factor (NGF) TrkA receptor [23], and TOLL-like receptors (TLRs) [24-27]

Line 102: Please add a comma after “pathway, but”.

Authors response: DONE

Line 131: Please add information about MiaPaCa-2 cells.

Authors response: We did not use MiaPaCa-2 cells in these studies.

Line 142: What percentage of saline was used?

Authors response: 0.9% normal saline- added in text

Line 143: Was the PBS used a 1x concentration?

Authors response:  Yes, 1x PBS - added in text  

Line 144: What percentage of saline was used?

Authors response: 0.9% normal saline - added

Line 151: What is SPF? Please explain here.

Authors response: Specific Pathogen Free (SPF) facilities are designed to maintain rodents in an environment that is free of certain (Not all) infectious organisms that are pathogenic and/or capable of interfering with research objectives. These are NOT completely germfree (or gnotobiotic) animals.

Line 161: Please replace “PANC1” with “PANC-1”.

Authors response: DONE

Line 163: What percentage of saline was used?

Authors response: 0.9% normal saline - added

Figure 1A: Please add metrics or reference to Methods of “palpable tumor” on the figure (i.e., palpable tumor was measured at 100 mm3). Please change “sacked” to “sacrificed” or “euthanized”

Authors response: DONE

Figure 1B: Why was Gem increased from 30, 50 to 100 mg/kg? What is the rationale for treating the mice on certain days (55, 65, 77, etc.)?

Authors response: The reason why we added GEM 4 days after pump implantation is a mandated procedure from the Veterinarian so that animals would recover from the surgery. The purpose of boosting up the dosage of GEM was to monitor the animal health response as well as the tumor growth to the chemo drug following surgical implantation of the pump.

       We wanted to reach the 100 mg/kg dosage of GEM which would simulate the dosage used in clinical trials. In Fig1G, we have enlarged the image to see the numbers in the X-axis.

Figure 2: The data in this figure appears to indicate that the OP+ASA+Gem combination therapy was more effective at reducing tumor volumes over time (based on Y-axis values). There is apparent variability between each individual mouse, but I understand that based on the representative images shown and the quantitation that combination is a better option. The osmotic pump with the combination therapy is even more effective.

Authors response: Agree

Figure 4: The data presented is very convincing. I didn’t know of a pancreatic cancer marker named PDX-1 until now and this makes the study more interesting. The acronym PDX also made me think if the authors would consider using PDX from pancreatic cancer patients and the same therapeutic conditions in this study? This should not be added to this study, but in the future this would provide even more clinical relevance and justify the use of the pump with standard of care.

Authors response: Agree

Figure 5: I appreciate the authors’ inclusion of the PI3K/Akt/mTOR signaling because of the long-standing literature on pancreatic cancer regulation of the processes listed in Lines 484-487. A very important publication to consider adding to this paper is cited below.

Lee S-W, Zhang Y, Jung M, Cruz N, Alas B, Commisso C. 2019. EGFR-Pak signaling selectively regulates glutamine deprivation-induced macropinocytosis. Developmental Cell. 50:381-392.

Authors response: Agree. We cited the requested paper.

Since most figures focused on the tumor microenvironment, it is also important to recognize potential influence from the surrounding stroma of the pancreas in future studies. The influence of the stroma may regulate or even block therapeutic efficacy in the cells used to test the pump with OP, ASA, and Gem.

Authors response: Agree

Also, is there a way to analyze Neu-1 by Western blot or at transcriptional levels? I have not seen its expression here, or is it assessed by activity level? Or is Neu-1 assessed indirectly by other proteins/genes? I am not familiar with Neu-1 but I am wondering if the authors can show it’s activity or expression changing in the figures because I don’t see it yet.

Please improve quality of the images on the box and whiskers plots. When I zoom in to higher than 200%, the quality of the images goes down and makes it hard to read the data. Some of the figures have artifacts on the graphs that must be removed to improve the viewing quality of the data. Please improve the quality of the IHC images, it is hard to see the staining at higher than 100% zoom. Also consider adding arrows to the IHC images where you have positive staining if it difficult to see.

Authors response: Thank you for this comment. We have increased the images in some of the figures and checked for any artifacts.

Figure 6: Data is very convincing, no changes needed except for those listed to improve the image quality. Statistics for all figures are sound and relevant so far.

Authors response: Thank you for this comment.

Figure 7: Most of the markers for EMT and possible MET decreased with the pump and combination therapy, but how do the authors plan to address the significant increase in CXCR4 expression in vivo? CXCR4 is a well-known marker that is induced by SDF1alpha in a nuclear-exclusive manner (citation below). So, perhaps the CXCR4 staining is mostly nuclear? Or maybe there is a way to only stain the non-nuclear form of this protein to gain a better understanding. This marker may bring more criticism to this figure than it is worth presenting based on its variable regulatory patterns even in other cancer cells.

Don-Salu-Hewage AS, Chan SY, McAndrews KM, Chetram MA, Dawson MR, Bethea DA, Hinton CV. 2013. Cysteine (C)-x-C receptor 4 undergoes transportin 1-depedent nuclear localization and remains functional at the nucleus of metastatic prostate cancer cells. PLoS One. 8(2):e57194.

Authors response: Thank you for this comment. We have added in the results text the following: “Furthermore, PUMP (ASA) and INJ-treated tumors did have lower CXCR4 expression compared to the untreated CTRL and GEM-only tumors; however, this was lost in the PUMP(OP) and PUMP(ASA+OP) treated tumors (Figure 7L and P). Interestingly, Seeber et al. [65] reported on CXCR4 expression in patients with pancreatic ductal adenocarcinoma (PDAC) that high CXCR4 expression is associated with an improved survival and a pro-inflammatory phenotype that may identify a subset of tumors with greater responsiveness to immunotherapeutic approaches.(Journal of Clinical Oncology Volume 39, Issue 15_suppl , Meeting Abstract | 2021 ASCO Annual Meeting I). We have also cited the PlosOne article.

CD24 inhibition is too variable to be convincing. Please increase the sample size to make sure that the expression is truly increased.

Authors response: Thank you for this comment. We were also concerned with CD24 as well. However, we have performed the following: “The standard way to identify CSCs is the expression of these characteristic markers on the cell surface. High CD44 expression and low expression of CD24 (CD44+/CD24low) are marker characteristics. For example, breast tumors expressing CD44+/CD24low have been shown to exhibit enhanced invasion and metastasis [68,69]. As shown in Figure 6 I, the CD44/CD24 ratio expressed on INJ, PUMP (ASA), and PUMP (OP) treated tumor tissues was not significantly different from GEM-treated tumor tissue. In contrast, the ratio of CD44/CD24 expressed on CTRL tumor tissue was significantly high compared to the GEM group. Interestingly, the ratio of CD44/CD24 expressed on the PUMP (ASA+OP) treated tumors was significantly reduced compared to the GEM group. These data support the evidence that untreated tumors have an invasive and metastatic cancer stem cell characteristic property. The PUMP (ASA+OP) treated tumors significantly reduced invasive and metastatic cancer stem cell characteristics.

Discussion: Please consider reiterating the significance of the study with a brief summary. Please answer questions in a brief paragraph about why pancreatic cancer metastasis is a problem, why the current standard of care chemotherapy has lower efficacy in these patients, and how you addressed it. After reading through a large amount of data, the reader needs to be reacquainted with the scope of what you investigated. The first two paragraphs are out of order. Please discuss the data in the order that it is presented in the Results section to prevent confusing the reader. The data also would be more convincing if the authors repeated these experiments in the future with the seemingly more aggressive MiaPaCa-2 cells. I didn’t see data in this manuscript for these cells and I don’t see supplemental figures. In the authors’ previous work they showed that the triple combination therapy similarly regulated migration in both MiaPaCa-2 and PANC-1. Gem was more effective at reducing migration in PANC-1 compared to MiaPaCa-2 cells, but it may be beneficial to see if MiaPaCa-2 can be used as a tool to study the ‘inflection’ period that the authors discussed. The transition point where tumors are less responsive to therapy is important when presenting work that is clearly relevant to pancreatic cancer patients. Another point to look into is the ‘stiffness’ of the pancreatic tissue after the treatments. Several studies have indicated that this phenotype is correlated with therapy responsiveness and unfortunately resistant to therapy (citations below).

Nguyen AV, Nyberg KD, et al. 2018. Stiffness of pancreatic cancer cells is associated with increased invasive potential. Integr Biol (Camb). 8(12):1232-1245.

Kai F, Laklai H, Weaver V. Trends Cell Biol. 2016;26:1–12.

Authors response: Thank you for these comments.

  1. With the comment to use MiaPaCa-2 cells for in vivo studies, we have reported the proof-of-evidence for a therapeutic targeting of Neu1 with OP without chemotherapeutic drug in impeding human pancreatic cancer growth and metastatic spread in heterotopic xenografts of eGFP-MiaPaCa-2 tumors growing in RAGxCγ double mutant mice (Gilmour et al. 2013 Cellular Signalling 25-2587). Here, OP therapy at 100 mg/kg daily dosage intraperitoneally expectedly impeded human pancreatic tumor growth in a time-to-progression growth rate compared to the untreated cohort. Following OP treatment, there was no significant increase in the time-to-progression tumor growth rate compared to a significant tumor growth rate for the untreated cohort. To confirm these results, we also found at necropsy that there was a significant reduction in tumor size and tumor weights at day 47 post-implantation taken from the OP treated tumor-bearing mice compared to the untreated cohort. This dose regime of OP had no side effects as determined by body weight and body condition scoring. Also, OP-treated cohort exhibited a reduction of phosphorylation of EGFR-Tyr1173, Stat1-Tyr701, Akt-Thr308, PDGFRα-Tyr754 and NFκBp65-Ser311 but an increase in phospho-Smad2-Ser465/467 and -VEGFR2-Tyr1175 in the tumor lysates from the xenografts of human eGFP-MiaPaCa-2 tumor-bearing mice. The findings identified a novel promising alternate therapeutic treatment of human pancreatic cancer.
  2. We have unpublished data on MiaPaCa-2 in vivo studies using OP and chemotherapeutics like GEM and other. OP in combination with chemo drug reduced the “inflection” point, i.e., the transition point where tumors are less responsive to therapy. We also have unpublished data on in vivo studies with pancreatic cancer cells using OP in combination with standard chemo drugs with long-term survival of 240 days. The data reveal no tumor growth, no metastatic disease, and no adverse side effects. For the last 80 days of this study, the animals received no drug treatments.
  3. the ‘stiffness’ of the pancreatic tissue after the treatments is an excellent study to do. We have cited the suggested paper.
  4. We have highlighted the results in the first paragraph of the discussion as suggested.

Reviewer 3 Report

The authors present an amazing amount of data describing potential new cancer therapies I really liked the thoroughness of the study and thought that for a pre-clinical study, there was a lot of good information presented to help inform the clinical studies to follow this work. The only suggested correction that I had was that there were scale bars missing from some of the H&E staining images in Figure 1.

Author Response

Thank you for the comment. Scale bar was added.

Round 2

Reviewer 1 Report

The summary paragraph in Discussion section is a good improvement, and it provides a general outline of what is being done in the study. Some minor grammar mistakes are fixed and sentences are more concise. Major questions are all answered in the cover letter. For the concern about the control group, the authors sufficiently elaborate the reason why they did not perform PUMP (ASA+OP) without GEM group experiment. Overall, I think the study is valid after revision.

Minor issues:

1.     Figure 1A-G is cut off due to some format problem, please fix that.

2.     One page 10, Figure 2, there is still a “E” right next to Figure 2C and 2D, please delete that considering Figure 2E is at the bottom.